# Turnover number predictions for kinetically uncharacterized enzymes using machine and deep learning

Alexander Kroll[1], Yvan Rousset[1], Xiao-Pan Hu[1], Nina A. Liebrand[1] & Martin J. Lercher ⓘ [1]✉

The turnover number $k_{cat}$, a measure of enzyme efficiency, is central to understanding cellular physiology and resource allocation. As experimental $k_{cat}$ estimates are unavailable for the vast majority of enzymatic reactions, the development of accurate computational prediction methods is highly desirable. However, existing machine learning models are limited to a single, well-studied organism, or they provide inaccurate predictions except for enzymes that are highly similar to proteins in the training set. Here, we present TurNuP, a general and organism-independent model that successfully predicts turnover numbers for natural reactions of wild-type enzymes. We constructed model inputs by representing complete chemical reactions through differential reaction fingerprints and by representing enzymes through a modified and re-trained Transformer Network model for protein sequences. TurNuP outperforms previous models and generalizes well even to enzymes that are not similar to proteins in the training set. Parameterizing metabolic models with TurNuP-predicted $k_{cat}$ values leads to improved proteome allocation predictions. To provide a powerful and convenient tool for the study of molecular biochemistry and physiology, we implemented a TurNuP web server.

The turnover number $k_{cat}$ is the maximal rate at which one active site of an enzyme converts molecular substrates into products. $k_{cat}$ is a central parameter for quantitative studies of enzymatic activities, and is of key importance for understanding cellular metabolism, physiology, and resource allocation. In particular, comprehensive sets of $k_{cat}$ values are essential for metabolic models that consider the cost of producing or maintaining enzymes[1–9], a prerequisite for accurate simulations of cellular physiology and growth[10]. Currently, no high-throughput experimental assays exist for $k_{cat}$, and experiments are both time consuming and expensive. Thus, $k_{cat}$ estimates are unavailable for most reactions; even for *Escherichia coli*, arguably the biochemically best-characterized organism, in vitro $k_{cat}$ is known for only ~10% of all enzyme-catalyzed reactions[11]. In genome-scale kinetic models of cellular metabolism, this issue is typically addressed by either sampling missing $k_{cat}$ values or fitting them to large

datasets[7,8,12,13]. However, these techniques typically result in inaccurate results, and fitted $k_{cat}$ values bear little relationship to known in vitro estimates[7,12,13].

Davidi et al.[11] estimated $k_{cat}$ values for enzymes in *Escherichia coli* using computationally calculated reaction fluxes and proteomic measurements across 31 different growth conditions. Their approach leads to $k_{cat}$ estimates that are highly correlated with experimentally measured values ($r^2 = 0.62$ on $\log_{10}$-scale). However, this approach is limited to very-well studied organisms, and even for *E. coli*, $k_{cat}$ values for only 436 enzymes could be estimated in this way. Recent advances in artificial intelligence have put the computational prediction of unknown $k_{cat}$ values from in vitro training data into reach, and two recent publications have explored this possibility. Heckmann et al.[14] developed a $k_{cat}$ prediction model for enzymes in *E. coli*. The model relies on detailed, expert-crafted input features such as

[1]Institute for Computer Science and Department of Biology, Heinrich Heine University, D-40225 Düsseldorf, Germany. ✉e-mail: Martin.Lercher@hhu.de

enzyme active site properties, metabolite concentrations, experimental conditions, and reaction fluxes calculated through flux balance analysis (FBA)[15]. It achieved a coefficient of determination $R^2 \approx 0.34$ on an independent test set. However, the complete, detailed input information is only available for a small subset of enzymatic reactions even in *E. coli*, limiting the applicability of this approach. A deep learning model that requires less detailed input features, DLKcat, was recently developed by Li et al.[16]. DLKcat predicts $k_{cat}$ using information about the enzyme's amino acid sequence and about one of the reaction's substrates, ignoring other reaction details such as products and co-substrates. In practical applications, $k_{cat}$ predictions are most important when no experimental measurements for closely related enzymes are available, and hence general prediction models should generalize well to such cases. However, while DLKcat can in principle be applied to any enzymatic reaction, its predictions become poor for enzymes not similar to those in the training set (see below).

Here, we present a general machine and deep learning approach for predicting in vitro $k_{cat}$ values for natural reactions of wild-type enzymes. In contrast to previous approaches, we represent chemical reactions through numerical fingerprints that consider the complete set of substrates and products of a reaction. To capture the enzyme properties, we use fine-tuned state-of-the-art protein representations as additional model inputs (Fig. 1). We created these enzyme representations using Transformer Networks, deep neural networks for sequence processing, which were trained with millions of protein sequences[17]. It has been shown for various prediction tasks that Transformer Networks outperform protein representations created with convolutional neural networks (CNNs)[18,19], which were used in previous models for predicting enzyme turnover numbers[16].

Our resulting Turnover Number Prediction model– TurNuP– outperforms a simple similarity-based approach and both previous methods for predicting $k_{cat}$[14,16]. We show that TurNuP generalizes well even to enzymes with <40% sequence identity to proteins in the

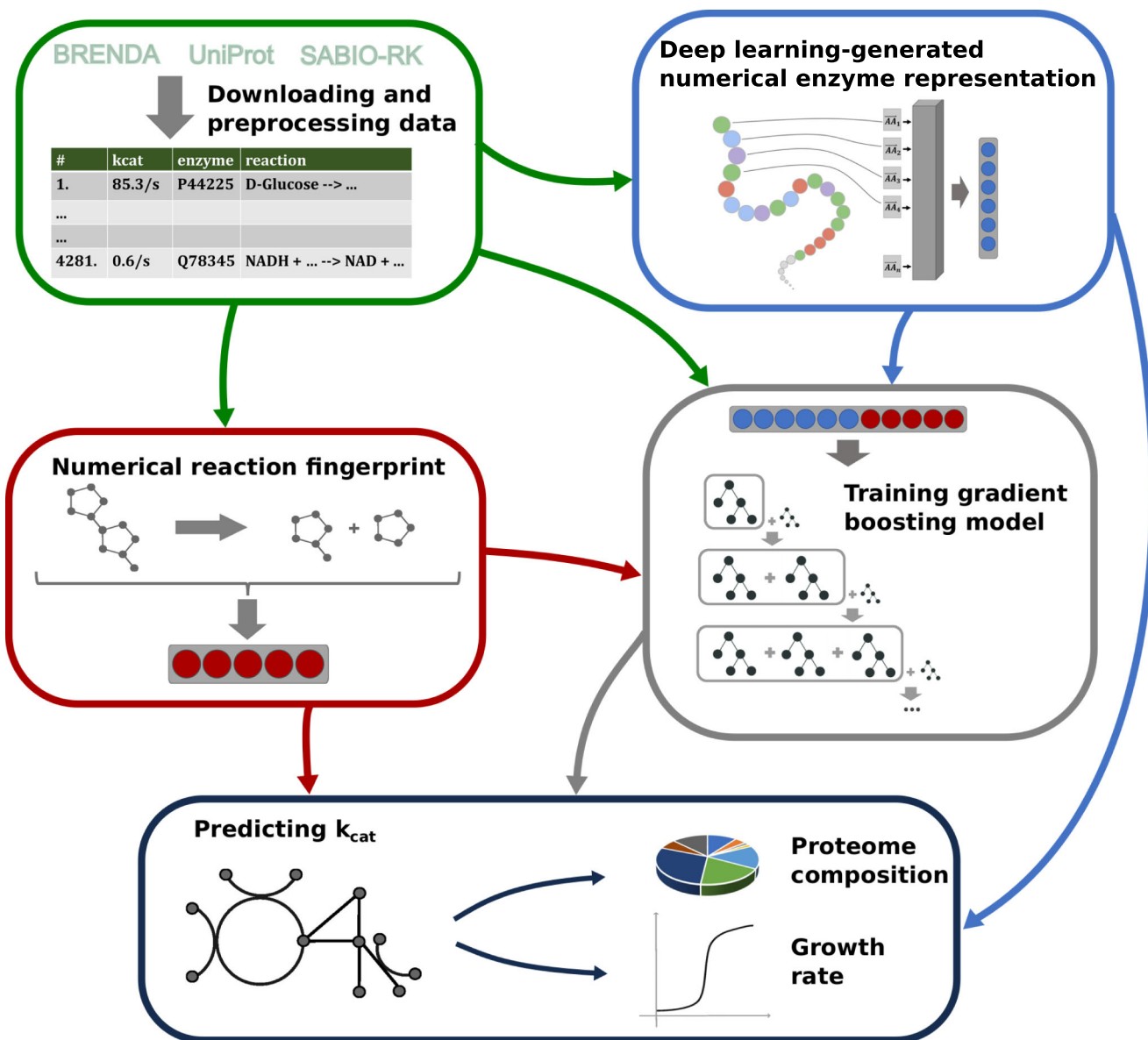

Fig. 1 | **Machine learning model to predict $k_{cat}$ from numerical enzyme representations and reaction fingerprints.** Experimentally measured $k_{cat}$ values are downloaded from three different databases. Enzyme information is represented with numerical vectors obtained from natural language processing (NLP) models that use the linear amino acid sequence as their input. Chemical reactions are represented using integer vectors. Concatenated enzyme-reaction representations are used to train a gradient boosting model to predict $k_{cat}$. After training, the fitted model can be used to parameterize metabolic networks with $k_{cat}$ values.

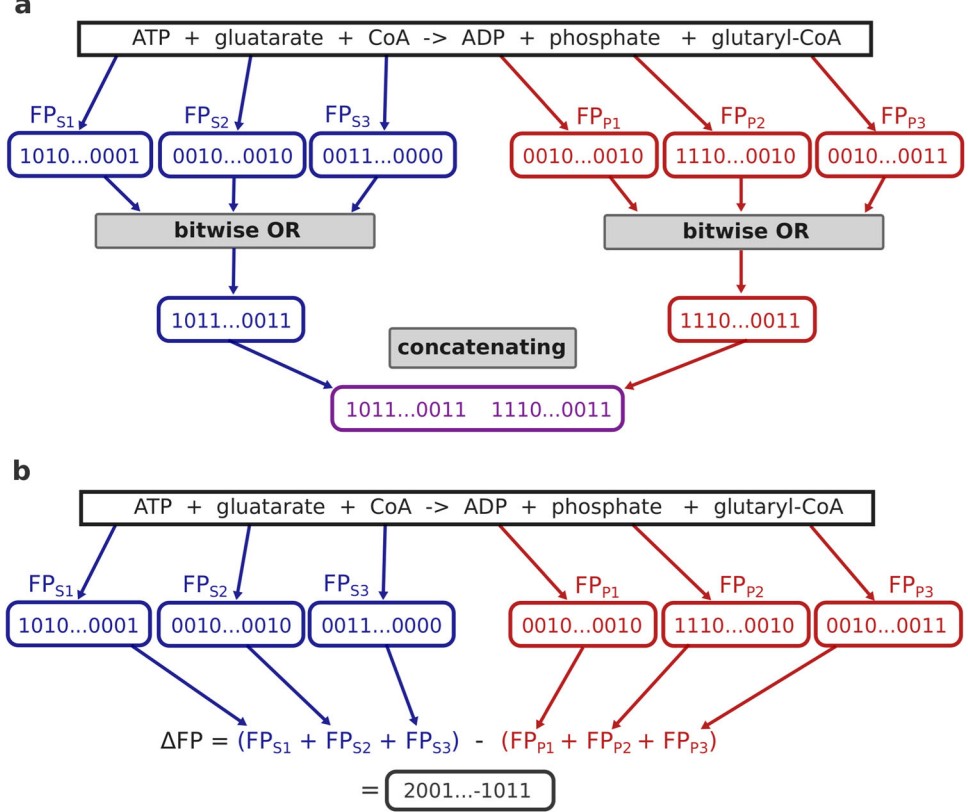

**Fig. 2 | Calculation of reaction fingerprints for an exemplary reaction.**
**a** Structural reaction fingerprints. Binary molecular fingerprints are calculated for each substrate and each product. The bitwise OR-function is applied to all substrates and also to all products. The resulting substrate and the resulting product vector are then concatenated. **b** Difference reaction fingerprints. Binary molecular fingerprints are calculated for each substrate and each product. All substrate fingerprint vectors are summed, and the same is done for all product fingerprint vectors. To create the difference fingerprint, the resulting product vector is subtracted from the substrate vector.

training set. Using genome-scale, enzyme-constrained metabolic models for different yeast species[16], we demonstrate that parameterizations with TurNuP $k_{cat}$ predictions lead to improved proteome allocation predictions. To facilitate widespread use of the TurNuP model, we not only provide a Python function for large-scale $k_{cat}$ calculations by bioinformaticians, but we also built an easy-to-use web server that requires no specialized software, available at https://turnup.cs.hhu.de.

## Results

### Obtaining training and test data
We compiled a dataset that connects $k_{cat}$ measurements with the corresponding enzyme sequences, reactant IDs, and reaction equations. The underlying data is derived from the three databases BRENDA[20], UniProt[21], and Sabio-RK[22]. Our aim was to build a turnover number prediction model for natural reactions of wild-type enzymes. We hypothesized that we do not have enough data to train a model to predict the catalytic effect of enzyme mutations or to predict the $k_{cat}$ value of non-natural enzyme-reaction pairs, which have not been shaped by natural selection. Hence, we removed all data points with non-wild-type enzymes and all non-natural reactions (see Methods, "Data preprocessing"). We removed redundancy by deleting data that was identical to other data points in the set, and we excluded data points with incomplete reaction or enzyme information. We also removed 55 outliers with unrealistically low or high measurements, i.e., reported $k_{cat}$ values that are either very close to zero ($<10^{-2.5}/s$) or that are unreasonably high ($>10^5/s$)[23]. If multiple different $k_{cat}$ values existed for the same enzyme-reaction pair, we took the geometric mean across these values.

This resulted in a final dataset with 4271 data points, comprising 2977 unique reactions and 2827 unique enzymes (for more details on data preprocessing, see Methods). We $\log_{10}$-transformed all $k_{cat}$ values to obtain a target variable with an approximate Gaussian distribution (Supplementary Fig. 1). We split the dataset into 80% training data and 20% test data in such a way that enzymes with the same amino acid sequence would not occur both in the training and in the test set. This splitting procedure does not prevent the inclusion of enzymes in the test set with close homologs in the training set. However, we found that only very few enzymes in the test set are highly similar to enzymes in the training set, i.e., only 70 out of 850 enzymes in the test set show ≥90% sequence identity to enzymes in the training set, and only 26 enzymes show ≥98% identity. Moreover, to evaluate model performance for different levels of enzyme similarities (see below), we divided the test set further into subsets with different levels of maximal sequence identities between training and test enzymes (Supplementary Fig. 2). To perform fivefold cross validations (CVs) for hyperparameter optimization of our machine learning models, we further split the training set into five disjoint subsets. The cross validation sets were constructed such that no two subsets contained enzymes with identical amino acid sequences, mirroring our strategy to split the total data into training and test sets.

### Numerical reaction fingerprints alone lead to reasonable $k_{cat}$ predictions
The $k_{cat}$ value of an enzyme-catalyzed reaction depends strongly on the catalyzing enzyme, but also on the chemical reaction itself. To integrate reaction information into our machine learning model, we used numerical reaction fingerprints. We compared the performance

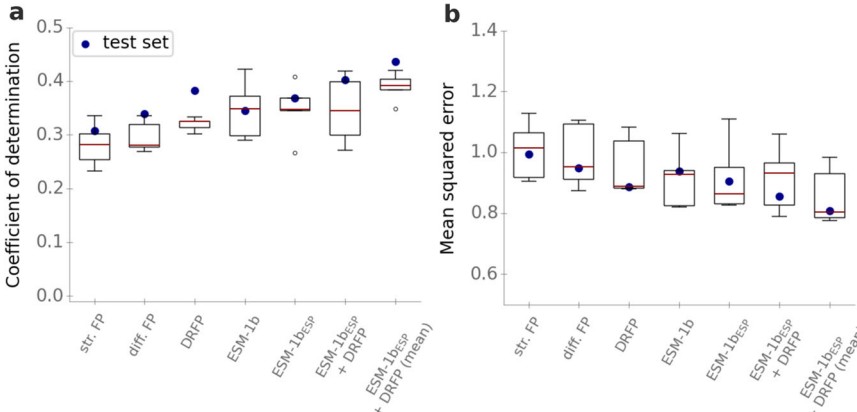

**Fig. 3 | Using enzyme and reaction information combined leads to improved $k_{cat}$ predictions. a** Coefficients of determination $R^2$ for models with different inputs. **b** Mean squared errors (MSE) on $\log_{10}$-scale. Boxplots summarize the results of the CV with $n = 5$ folds on the training set with the best sets of hyperparameters; blue dots show the results on the test set using the optimized models trained on the whole training set. We used a 2× interquartile range for the whiskers, the boxes extend from the lower to upper quartile values, and the red horizontal lines are displaying the median of the data points. Model performances are plotted for the models with structural reaction fingerprints (str. FP), difference reaction fingerprints (diff. FP), differential reaction fingerprints (DRFP), ESM-1b vectors (ESM-1b), task-specific ESM-1b vectors (ESM-1b$_{ESP}$), and with enzyme and reaction information (ESM-1b$_{ESP}$ + DRFP). Source data are provided as a Source Data file.

of three different types of such representations: structural, difference, and differential reaction fingerprints.

To create structural reaction fingerprints, one first calculates a 1638-dimensional binary molecular fingerprint for each substrate and each product, designed to encode structural information of small molecules; e.g., entries of these fingerprints can encode if a certain substructure is present in a molecule. The bitwise OR-function is then applied to all substrate fingerprints and separately to all product fingerprints, resulting in two 1638-dimensional binary vectors with molecular information about the substrates and about the products, respectively. These two vectors are concatenated, providing a 3276-dimensional binary vector with structural information about the reaction[24] (Fig. 2a).

The calculation of difference reaction fingerprints starts with a different, 2048-dimensional binary fingerprint for each substrate and each product. All substrate fingerprint vectors are summed to provide a single substrate vector, and all product fingerprint vectors are summed to provide a single product vector. This product fingerprint is then subtracted from the substrate fingerprint, resulting in a 2 048-dimensional reaction fingerprint with positive and negative integers[25] (Fig. 2b).

The differential reaction fingerprints (DRFPs) are calculated in a similar fashion compared to the two fingerprints described above. However, instead of first calculating fingerprints for each reactant and then combining these into a single reaction vector, a fingerprint for the whole reaction is directly created. To achieve this, substructures of all substrates and products are identified. All substructures that are only present either in the substrates or the products are then mapped to a 2048-dimensional binary fingerprint using hash-functions[26].

Creating the three reaction fingerprints described above can result in information loss. For example, if multiple substrates or products are present, it is not possible to distinguish between properties of different reactants. However, the fingerprints allow to store information about reactions in a fixed-length vector even for varying numbers of reactants. Storing every reactant in a separate fingerprint would result in much larger input vectors for reactions with multiple substrates and products. Moreover, the resulting vectors would not be invariant to the order of the substrates and products and would vary in length for varying numbers of reactants.

To test how well the reaction fingerprints alone can predict the turnover numbers of enzyme-catalyzed reactions, we trained two gradient-boosting models to predict $k_{cat}$, each with one of the reaction fingerprints as the only input. Gradient boosting models consist of many decision trees that are built iteratively during the training process. In the first iteration, a single decision tree is built that tries to predict the correct $k_{cat}$ for all data points in the training set. In all following iterations, a new decision tree is built in order to reduce the errors that have been made by the already existing trees. After training, many different decision trees exist that ideally focus on different aspects of the input features and that try to predict the correct outcome as an ensemble[27].

We performed a fivefold CV with a random grid search for hyperparameter optimization for all three models. After hyperparameter optimization, we chose the set of hyperparameters with the highest coefficient of determination $R^2$ across CV sets, and we re-trained each model with its best hyperparameters on the whole training set. On the test set, the resulting model with structural reaction fingerprints as its inputs achieves a coefficient of determination $R^2 = 0.31$, a mean squared error MSE = 0.99, and a Pearson correlation coefficient $r = 0.56$ on the test set. The model with difference reaction fingerprints achieves slightly improved results, with $R^2 = 0.34$, MSE = 0.95, and $r = 0.60$ on the test set. The best model performance is achieved using DRFPs, with $R^2 = 0.38$, MSE = 0.89, and $r = 0.62$ on the test set (Fig. 3). Thus, a model based on chemical reaction information alone can already predict about a third of the variation in $k_{cat}$ across enzyme-catalyzed reactions.

As the DRFPs led to improved performance on the test set and during CV (Fig. 3), we chose DRFPs to represent the catalyzed chemical reactions in the further analyses. To test if the better performance of the DRFPs is statistically significant, we used a two-sided Wilcoxon signed-rank test that compared the absolute errors of the models on the test set, resulting in $p = 0.064$ (DRFP vs. difference fingerprints) and $p = 2.61 \times 10^{-4}$ (DRFP vs. structural fingerprint). Thus, while the difference in absolute errors between DRFPs and difference reaction fingerprints is not statistically significant at the commonly used 5% level, a value close to 0.05 indicates that the DRFPs indeed likely lead to improved model performance.

## Numerical enzyme representations alone lead to reasonable $k_{cat}$ predictions

The turnover number $k_{cat}$ of an enzyme-catalyzed reaction is highly dependent on the catalyzing enzyme. It can vary by orders of magnitude even between isoenzymes that catalyze the same reaction but differ in amino acid sequence[28]. To account for this dependence when predicting $k_{cat}$, it is crucial to create meaningful enzyme representations as inputs

to machine learning models. In recent years, deep learning architectures that were originally developed for natural language processing (NLP) tasks, such as translating a sentence from one language into another, have been applied successfully to the creation of numerical protein representations from amino acid sequences[17,29]. When applied to natural languages, NLP models typically represent all words in a sentence through numerical vectors that encode information about the words' contents and positions. When applying NLP models to protein sequences, proteins replace sentences and amino acids replace words.

The current state-of-the-art architecture for NLP tasks is a Transformer Network[30], which can, in contrast to previous methods, process all words of a sequence with arbitrary length simultaneously. The Facebook AI Research team trained such a Transformer Network, called ESM-1b, with a dataset of ~27 million protein sequences from the UniRef50 dataset[31] to create 1280-dimensional numerical protein vectors. The ESM-1b model was trained in a self-supervised fashion, i.e., 10–15% of the amino acids in a sequence were masked at random, and the model was trained to predict the identity of the masked amino acids (Supplementary Fig. 3a). It has been shown that the resulting representations contain rich information about the structure and the function of the proteins[17,32,33]. Using the pre-trained ESM-1b model[17], we calculated these 1280-dimensional representations for all enzymes in our dataset, in the following referred to as ESM-1b vectors.

In a previous project[32], we created a fine-tuned and task-specific version of the ESM-1b model that led to improved predictions for the substrate scope of enzymes, a problem for which abundant training data exists (Supplementary Fig. 3b). Such comprehensive data is required to re-train the ESM-1b model, but is not available for $k_{cat}$, and we were thus unable to create a version specific to the task of predicting $k_{cat}$. However, we speculated that the ESM-1b vectors fine-tuned previously for the prediction of enzyme-substrate pairs might also improve $k_{cat}$ predictions. To test this hypothesis, we used our previously published model, ESP[32], to calculate fine-tuned representations for all enzymes in our dataset. In the following, we will refer to these representations as ESM-1b$_{ESP}$ vectors.

We tested how well models that use enzyme information alone can predict turnover numbers. We trained a gradient boosting model[34] that used either the ESM-1b or ESM-1b$_{ESP}$ vectors to predict the $k_{cat}$ value of enzyme-catalyzed reactions, without using any additional information on the reaction or on substrates or products.

To optimize the hyperparameters of the gradient boosting models, we again performed fivefold cross validations (CV) with a random grid search on the training set. Afterwards, we re-trained each model with its best hyperparameters on the whole training set. On the test set, the model with ESM-1b vectors as its input achieves a coefficient of determination $R^2 = 0.36$, a mean squared error MSE = 0.92, and a Pearson correlation coefficient $r = 0.60$ (Fig. 3). The model with ESM-1b$_{ESP}$ vectors achieves slightly improved performance, with $R^2 = 0.37$, MSE = 0.91, and $r = 0.61$ on the test set (Fig. 3). Since the ESM-1b$_{ESP}$ vectors lead to improved performance on the test set and during CV, we chose to represent enzymes through ESM-1b$_{ESP}$ vectors in the following. However, as model performance is quite similar for both enzyme representations, we tested if the difference in model performance is statistically significant. We used a one-sided Wilcoxon signed-rank test that compared the absolute errors made by both models on the test set, resulting in $p = 0.41$. Thus, the difference in absolute errors is not statistically significant at the commonly used 5% level. This finding indicates that the observed performance improvement with ESM-1b$_{ESP}$ vectors might be due to random effects, and that we cannot be sure that the model with ESM-1b$_{ESP}$ vectors is indeed superior.

## A joint model with enzyme and reaction information leads to improved $k_{cat}$ predictions

To train a Turnover Number Prediction model (TurNuP) with enzyme and reaction information, we concatenated the ESM-1b$_{ESP}$ vector and

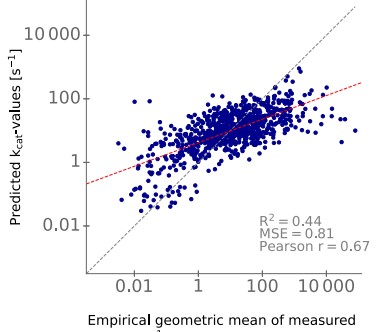

**Fig. 4 | Comparison of predicted and experimentally measured $k_{cat}$ values.** $k_{cat}$ values predicted with the complete TurNuP model, plotted against the corresponding experimental measurements. Each dot is one data point from the test set. The gray dashed line is a diagonal line that indicates perfect correlation. The red dashed line displays the true correlation between predicted and measured values. Source data are provided as a Source Data file.

the DRFPs for every data point in our dataset. We used this resulting vector as the input for a gradient-boosting model. As before, we performed a fivefold CV with a random grid search for hyperparameter optimization, trained the model with the best set of hyperparameters on the whole training set, and validated it on the test set. This model achieves a coefficient of determination $R^2 = 0.40$, a mean squared error MSE = 0.86, and a Pearson correlation coefficient $r = 0.64$ on the test set.

We found that calculating the mean of the predictions provided by the models that only use ESM-1b$_{ESP}$ vectors and DRFPs as their model input leads to increased model performance during CV and on the test set. This final TurNuP model achieves a coefficient of determination $R^2 = 0.44$, a mean squared error MSE = 0.81, and a Pearson correlation coefficient $r = 0.67$ on the test set (Figs. 3 and 4). We hypothesize that the mean of two separate models leads to improved performance, because the information contained in the ESM-1b$_{ESP}$ vectors and the DRFPs is partly redundant. Training a joint model allows the model to focus only on one of the input vectors when extracting redundant information. However, when this information is provided into separate models through separate input representations, the information is extracted from both representations, which can lead to more robust predictions.

Using enzyme and reaction information improves performance compared to using only enzyme or only reaction information (Fig. 3). To compare these differences statistically, we used a one-sided Wilcoxon signed-rank test, testing if the absolute errors on the test set for the joint model are lower than for the models with either only enzyme or only reaction information. These tests showed that the differences are statistically significant at the 5% level, with $p = 0.0049$ (DRFP) and $p = 1.2 \times 10^{-7}$ (ESM-1b$_{ESP}$). However, the improvement for the joint model is relatively small, indicating that the information stored in the reaction fingerprints and in the enzyme representations are overlapping. This overlap is not surprising, as the enzyme sequence contains information about the catalyzed reaction[32]; conversely, given that enzymes evolve on fitness landscapes shaped by the catalyzed reactions, the chemical reaction likely also contains information about the type of catalyzing enzyme.

TurNuP achieves a mean absolute deviation of predicted from experimental $k_{cat}$ values of 0.69 on a $\log_{10}$-scale, which means that predictions and measured values deviate on average by 4.8-fold. Plotting the correlation between different experimental measurements for the same enzyme-reaction pair (Supplementary Fig. 4) shows that there is substantial variance even between measurements for the same $k_{cat}$ value. This noise in the training and test data indicates

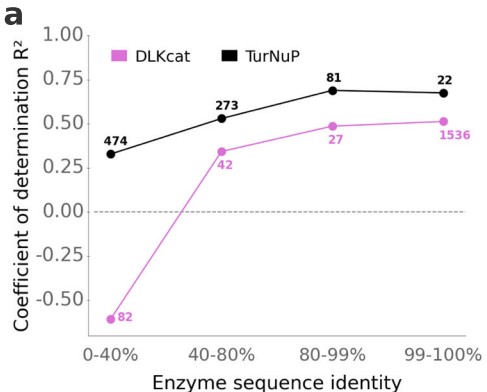

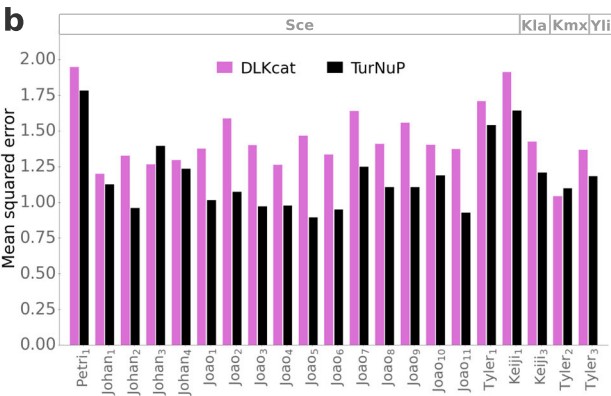

**Fig. 5 | TurNuP predictions are more accurate for enzymes similar to proteins in the training set and outperform an existing deep learning model. a** Coefficients of determination $R^2$ for the test sets for our TurNuP model (black) and the previously published DLKcat model[16] (magenta) for different levels of maximal enzyme sequence identity compared to enzymes in the training set. Numbers next to points show how many data points of this category are in the test set. The horizontal dashed line corresponds to a model that predicts the same mean $k_{cat}$ value for all test data points. **b** Mean squared errors (MSE) for the prediction of absolute proteome data compared to experimental data. Proteome predictions were achieved with enzyme-constrained genome-scale models, parameterized with $k_{cat}$ values predicted with TurNuP (black) or with the DLKcat model (magenta). Proteome data was predicted for four different yeast species (Sce, *Saccharomyces cerevisiae*; Kla, *Kluyveromyces lactis*; Kmx, *Kluyveromyces marxianus*; Yli, *Yarrowia lipolytica*) in 21 different culture conditions (see Methods for details). Source data are provided as a Source Data file.

that it is difficult to develop a prediction model achieving much better accuracy unless less noisy data becomes available.

Figure 4 shows that TurNuP tends to systematically overestimate extremely low and underestimate very high $k_{cat}$ values. This phenomenon likely arises due to a common statistical effect known as regression dilution: noise present in the input features leads to a flattening of the slope of the line that describes the correlation between the predicted and true values for the target variable[35]. While the chosen input features in TurNuP can account for ~44% of the variance in $k_{cat}$, they only approximately capture the true, unknown determinants of $k_{cat}$. The input features can thus be considered noisy representations of those true determinants, leading to regression dilution. We further hypothesize that non-optimal experimental conditions (such as pH, temperature, missing metal ions) play an important factor in cases where very low $k_{cat}$ values were observed but higher $k_{cat}$ values were predicted. Supplementary Fig. 4, which displays the similarity between different measurements for the same enzyme-reaction pairs, supports this hypothesis: a large variance is present especially for very low $k_{cat}$ measurements. Unfortunately, we could not include reaction conditions as input features to overcome this issue, since experimental conditions are not available for most data points in the enzyme databases used.

To compare the gradient boosting model to alternative machine learning models, we also trained a linear regression model, a random forest model, and a fully connected neural network for the task of predicting $k_{cat}$ values from the combined ESM-1b$_{ESP}$ and DRFP vectors. However, these models performed worse compared to the gradient boosting model (Supplementary Table 1). To test if the better performance of the gradient boosting model is statistically significant, we used a one-sided Wilcoxon signed-rank test that compared the absolute errors of the models on the test set, resulting in $p = 4.29 \times 10^{-11}$ (gradient boosting vs. linear regression), $p = 3.38 \times 10^{-7}$ (gradient boosting vs. random forest), and $p = 8.97 \times 10^{-4}$ (gradient boosting vs. neural network). Thus, the additional machine learning models indeed lead to statistically significant worse results compared to the gradient boosting model.

It is noteworthy that the accuracies on the test set are partly higher than the accuracies achieved during CV (Fig. 3a). To calculate the results of the CV, the model is validated on data that is originally part of our training data set and has been used for hyperparameter optimization, but it has not been used to train the model during this round of CV. The improved performance on the test set may result from the fact that before validation on the test set, models are trained with approximately 700 more samples than before each cross-validation; the number of training samples likely has a substantial influence on model performance.

## TurNuP provides meaningful predictions even if no close homologs with known $k_{cat}$ exist

In our study on predicting the substrate scope of enzymes[32], we found that prediction performance depends strongly on the sequence similarity between a target enzyme and enzymes in the training set, consistent with the widely held belief that enzymes are more likely to be functionally similar if they have more similar sequences[36]. We hence examined the performance of TurNuP for enzyme sets that differed in their maximal similarity to proteins in the training set. We partitioned the enzymes in the test set based on their maximal sequence identity to enzymes in the test set, resulting in four subsets with 0–40%, 40–80%, 80–99%, and 99–100% maximal sequence identity, respectively. We calculated TurNuP's coefficient of determination for all four categories (Fig. 5a, black points). As expected, prediction performance decreases with increasing distance of the enzyme's amino acid sequence to proteins in the training set. While TurNuP's coefficient of determination is $R^2 = 0.67$ for 99–100% sequence identity, it decreases to $R^2 = 0.33$ for enzymes with a maximal sequence identity below 40%.

A simple, straight-forward, and often used alternative method to predict approximate $k_{cat}$ values is to simply average over the $k_{cat}$ values of the most similar enzymes. Such simple averages are expected to work well in cases where kinetically characterized homologs with highly similar amino acid sequences exist; in contrast, they are unlikely to provide good estimates if no close homologs with known $k_{cat}$ exist. As expected, for enzymes in the test set with close homologs in the training set (99–100% max. identity), the geometric mean across the three most similar enzymes in the test set leads to reasonable estimates, with $R^2 = 0.21$ ($N = 22$). In contrast, averaging over the three most similar enzymes leads to a dismal $R^2 = 0.02$ if no close homologs exist in the training data (0–40% max. identity, $N = 474$). These results demonstrate that any sophisticated prediction model for turnover numbers will be most relevant for enzymes for which no close homologs with known $k_{cat}$ exist. As expected, TurNuP predictions are statistically significantly better than those provided by simple averages, across the complete test set ($R^2 = 0.44$ vs. $R^2 = 0.24$, $N = 851$,

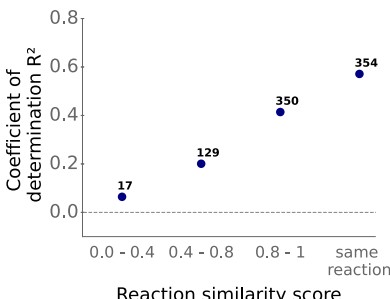

**Fig. 6 | TurNuP makes good predictions even for unseen reactions.** Coefficients of determination $R^2$, calculated separately on different splits of the test set. The test set was split according to the maximal reaction similarities compared to reactions in the training set. Source data are provided as a Source Data file.

$p = 0.013$ from one-sided Wilcoxon signed-rank test) as well as for the similarity classes (0–40% max. identity: $p = 3.4 \times 10^{-9}$; 99–100% max. identity: $p = 0.0023$).

## Good predictions even for unseen reactions

In the previous subsection, we showed that model performance is highest for enzymes that are similar to proteins in the training set. Similarly, it appears likely that the model performs better when making predictions for chemical reactions that are also in the training set. To test this hypothesis, we divided the test set into data points with reactions that occurred in the training set ($N = 354$) and those with reactions that did not occur in the training set ($N = 496$).

Unsurprisingly, TurNuP performs better for those data points with reactions that occurred in the training set, achieving a coefficient of determination $R^2 = 0.57$, a mean squared error MSE = 0.51, and a Pearson correlation coefficient $r = 0.78$ (Fig. 6). However, for those test data points with reactions that TurNuP has not seen during training, model performance is still good, resulting in a coefficient of determination $R^2 = 0.35$, mean squared error MSE = 1.02, and Pearson correlation coefficient $r = 0.60$.

For those test data points with reactions not present in the training set, we wondered if a high similarity of the reaction compared to at least one reaction in the training set leads to improved predictions, analogous to what we observed for enzymes with higher sequence identities. For each reaction not present in the training set, we calculated a maximal pairwise similarity score compared to all reactions in the training set based on their structural reaction fingerprints. We indeed found that prediction performance is higher for those data points with reactions more related to training reactions (Fig. 6). Even for reactions that are not highly similar to training reactions, i.e., reactions with a similarity score between 0.4 and 0.8, prediction accuracy is still moderate with a coefficient of determination $R^2 = 0.20$, a mean squared error MSE = 1.30, and a Pearson correlation coefficient $r = 0.47$. Only for those 17 test data points with reactions that share almost no similarity compared to training reactions, model performance is low, achieving a coefficient of determination $R^2 = 0.06$, a mean squared error MSE = 1.26, and a Pearson correlation coefficient $r = 0.29$. We conclude that TurNuP can provide useful predictions for most unseen reactions even if these are not highly similar to previously seen reactions.

## TurNuP outperforms previous models for predicting $k_{cat}$

Heckmann et al.[14] trained and validated a machine learning model for the prediction of $k_{cat}$ values for *E. coli*. As enzyme-related input features, their model used enzyme molecular weight and global structural disorder, as well as several molecular details of the active site: number of residues, solvent access, depth, hydrophobicity, secondary structure, and exposure. Additional input features were reaction flux,

number of substrates, the dissociation constant $K_M$, EC number, substrate and product concentration, thermodynamic efficiency, and the pH value and temperature at which $k_{cat}$ was measured in vitro. Out of this large set of features, the most important input was found to be the reaction flux, which was calculated by performing parsimonious flux balance analyses (pFBA)[37,38]. The total number of training and validation data points was limited to 215, as Heckmann et al.[14] only considered reactions from *E. coli*, and as many input features are not available for most enzymes—the least widely available features were information about the enzymes' active site, and the pH and temperature of the in vitro experiment. The model achieved a coefficient of determination $R^2 \approx 0.34$ on a test set. Since Heckmann et al.'s test dataset is small and only contains measurements for a subset of reactions from a single organism, it is not possible to directly compare the performance of their model to TurNuP's performance, which was evaluated on a much more extensive and general test dataset.

The DLKcat model by Li et al.[16] examined the same problem addressed here, the prediction of $k_{cat}$ values across the space of all possible enzymatic reactions. Therefore, we undertook a more in-depth comparison to DLKcat. To predict $k_{cat}$ values, DLKcat uses information extracted from the amino acid sequence and from one of the substrates of the reaction. Instead of using state-of-the-art methods for encoding protein information, i.e., transformer networks applied to protein amino acid sequences, convolutional neural networks (CNNs) were applied. Moreover, since DLKcat only uses information about one of the substrates, important information about additional substrates and the products can be missing. The DLKcat test set was constructed as a random subset of the full data. In comparison, our own test set was constructed such that it contained no enzymes with sequences that are identical to those of the training set, i.e., it is biased towards enzymes that are distinct from those used for training. This intentional bias, which was introduced to assess the ability of TurNuP to extrapolate to new enzymes, means that a direct performance comparison between DLKcat and TurNuP on their respective test sets is not meaningful. To account for this bias, we divided both test sets—those of TurNuP and of DLKcat—into different splits according to the maximal enzyme sequence identity compared to the respective training enzymes (Fig. S2). Given that both models were developed for the same purpose— predicting $k_{cat}$ for the reactions catalyzed by arbitrary enzymes—and were evaluated on large and diverse test data, a performance comparison on these sequence identity splits is meaningful.

Figure 5a shows that TurNuP (black) achieves substantially higher coefficients of determination than DLKcat (magenta) for all categories of enzyme sequence identity. Comparing the distributions of absolute errors of the two models, we find that the superior performance of TurNuP is statistically significant for all subsets ($p = 4.3 \times 10^{-11}$ (0-40%), $p = 0.0017$ (40–80%), $p = 0.00037$ (80–99%), and $p = 0.048$ (99–100%); one-sided Wilcoxon-Mann-Whitney tests). Although TurNuP performs better than DLKcat in each of the four categories of enzyme sequence identities (Fig. 5a), DLKcat achieves the same $R^2$ value ($R^2 = 0.44$) on its overall test set, compared to the TurNuP model on its overall test set ($R^2 = 0.44$). This counter-intuitive observation is an example of Simpson's paradox. It is caused by the differential distribution of data points across categories in the DLKcat and TurNuP test sets. 91% of the data points in the DLKcat test set fall into the 99–100% identity class (see the numbers in Fig. 5a), while the majority of data points in the TurNuP test set (56%) have <40% sequence identity to any enzymes in the corresponding training set and are hence much harder to predict. In contrast to TurNuP, DLKcat was not challenged during its training to predict $k_{cat}$ values for enzymes with dissimilar amino acid sequences, likely explaining its poor performance for enzymes without close homologs in the training set.

Li et al.[16] designed a pipeline to use predicted $k_{cat}$ values for the parameterization of enzyme-constrained genome-scale metabolic

models, with the goal of predicting the proteome allocation patterns of yeast species. They compared the resulting proteome predictions to absolute proteomics measurements for four yeast species in 21 different environments. We employed this pipeline to test if $k_{cat}$ values calculated with the TurNuP model lead to improved proteome predictions. In 19 out of 21 environment-species combinations, our $k_{cat}$ values led to improved predictions ($p = 0.00010$, one-sided binomial test). The mean squared errors between measured and predicted protein abundances improved on average by ~18% when using TurNuP (Fig. 5b).

## Using additional input features does not improve model performance

TurNuP employs very general input features, using only the enzyme's linear amino acid sequence and information on the reaction's substrates and products. However, it is unclear if these features cover all important aspects for predicting $k_{cat}$. To test if we can improve prediction quality, we examined three potential additional input features: Michaelis constants $K_M$, Codon Adaptation Indices (CAIs), and reaction fluxes.

The Michaelis constant $K_M$ is defined as the substrate concentration at which an enzyme works at half of its maximal catalytic rate; hence, $K_M$ quantifies the affinity of an enzyme for its substrate. It has been shown that $k_{cat}$ is correlated with the enzyme's Michaelis constant(s) of the reaction's substrate(s)[23]. To utilize this correlation for the prediction of $k_{cat}$, we determined $K_M$ values for all enzyme-substrate combinations in our dataset. Where available, we extracted suitable $K_M$ values from the BRENDA database[20] (~7% of the enzyme-substrate pairs in our dataset); for all other cases we applied a machine learning model that uses numerical representations of the substrate and the enzyme as its input to predict $K_M$[39]. For reactions with multiple substrates, we took the geometric mean of all $K_M$ values to obtain a single $K_M$ value for every data point. To calculate how much variance of $k_{cat}$ can be explained by $K_M$, we fitted a linear regression model to the training set, with the $\log_{10}$-transformed $K_M$ value as the only input. The linear regression model achieves a coefficient of determination $R^2 = 0.11$, a mean squared error MSE = 1.28, and a Pearson correlation coefficient $r = 0.34$ on the test set (Supplementary Fig. 5). We thus considered $K_M$ a promising candidate for improving the TurNuP predictions.

The second additional input feature, the Codon Adaptation Index (CAI), quantifies the synonymous codon usage bias of protein-coding genes. It is widely used as an indicator of gene expression and protein levels, with highly expressed genes typically using more 'preferred' codons than less highly expressed genes[40]. The CAI is a value between 0 and 1 that describes the similarity of synonymous codons frequencies between a given gene and a set of highly expressed genes, where values close to 1 indicate nearly optimal codon usage, typically associated with a high expression level in evolutionarily relevant environments. We calculated the CAI for all enzymes in our dataset originating from *E. coli*. We fitted a linear regression model to the corresponding 237 data points in the training set, with CAI as the only input feature. We validated the model on 66 test data points (Supplementary Fig. 6). The model achieved a coefficient of determination $R^2 = 0.012$, a MSE = 1.31, and a Pearson correlation coefficient $r = 0.12$ on the test set, indicating that CAI cannot explain much of the variance of $k_{cat}$ values. Hence, we did not consider CAI a promising candidate for improving the TurNuP predictions, and we did not calculate CAI for other organisms beyond *E. coli*.

The most important input feature in the $k_{cat}$ prediction model established by Heckmann et al. for reactions in *E. coli*[14] was an estimate of the reaction flux, calculated using parsimonious flux balance analysis (pFBA)[37,38] across a broad range of nutrient conditions. For 108 metabolic genome scale models from the BiGG database[41], we calculated fluxes in a similar way as Heckmann et al. (Methods). For further

analyses, we selected the six BiGG models of distinct species that showed the highest Pearson correlation between predicted fluxes and measured $k_{cat}$ values in the training set. We mapped the calculated fluxes to $k_{cat}$ values from our dataset. In cases where no metabolic genome scale model was available for an organism, we mapped the flux of an identical reaction but from a different organism to the data point. If we were not able to find the identical reaction in the BiGG database, we selected the most similar one using a similarity score (see Methods). To calculate how much variance of the $k_{cat}$ values can be explained by the calculated fluxes, we fitted a linear regression model to the training set, with the $\log_{10}$-transformed fluxes as the only input. The fitted model achieves a coefficient of determination $R^2 = 0.021$, a MSE = 1.40, and a Pearson correlation coefficient $r = 0.15$ on the test set (Supplementary Fig. 7). Thus, we found no evidence for a high predictive power of fluxes beyond *E. coli*; however, as fluxes were the most important predictor for $k_{cat}$ in ref. 14, we still retained them as a potential additional input feature for TurNuP.

To test if adding $K_M$ and reaction flux as input features improves model performance, we trained a new model. As the model input, we created a concatenated vector comprised of the enzyme ESM-1b$_{ESP}$ vector, the differential reaction fingerprint, the reaction flux, and the Michaelis constant $K_M$ for every data point. For a gradient boosting model, we then performed a fivefold CV with a random grid search for hyperparameter optimization. Afterwards, we trained the model with the best set of hyperparameters on the complete training set. On the test set, this model achieves a coefficient of determination $R^2 = 0.39$, a MSE = 0.87, and a Pearson correlation coefficient $r = 0.63$. Thus, model performance did not improve compared to the model without the additional input features flux and $K_M$.

This finding does not preclude the usefulness of flux and $K_M$ for predicting $k_{cat}$. On the one hand, the result suggests that the enzyme and reaction representations already contain the relevant information offered by the additional input features used here. On the other hand, it is possible that accurate measurements of flux and $K_M$, as opposed to the approximate predictions employed here, could enhance the predictive power of these variables for $k_{cat}$.

## The TurNuP web server provides an easy access to the prediction model

We implemented a web server that facilitates an easy use of the TurNuP model without requiring programming skills or the installation of specialized software. It is available at https://turnup.cs.hhu.de. As input, the web server requires an enzyme amino acid sequence and representations of all substrates and all products; the latter can be provided either as SMILES strings, KEGG Compound IDs, or InChI strings. Users can enter a single enzymatic reaction into an online form, or upload a CSV file with multiple reactions. Since TurNuP was trained only with natural reactions of wild-type enzymes, we recommend to use the web server only for such enzyme-reaction pairs. Additionally to a $k_{cat}$ prediction, the web server outputs the maximal enzyme sequence identity compared to all training enzymes and a maximal reaction similarity score compared to all training reactions. As discussed above, higher scores indicate higher prediction performance, and thus, these scores can be used to estimate the accuracy of the $k_{cat}$ prediction.

## Discussion

Predicting the turnover number of enzyme-catalyzed reactions is a complex task, and the available datasets for model training are small and noisy. For example, Bar-Even et al.[23] found that up to 20% of the entries in BRENDA differ from the entries in the reference papers, probably caused by copying errors and erroneous replacements of units. Even aside from such obvious errors, the variance of $k_{cat}$ measurements for the same enzyme-reaction pairs between different studies can be high. We found an average deviation of 5.7-fold (mean

deviation on $\log_{10}$-scale = 0.75) between two $k_{cat}$ measurements for the same enzyme-reaction pair (Supplementary Fig. 4). This variance is likely not only due to errors in the databases, but also to different experimental procedures or varying assay conditions, such as temperature and pH value. When comparing a single measurement to the geometric mean of all other measurements for the same enzyme-reaction pair, we found an average deviation of 3.3-fold (mean deviation on $\log_{10}$-scale = 0.52). This compares to an average deviation of 4.8-fold (mean deviation on $\log_{10}$-scale = 0.69) of TurNuP $k_{cat}$ predictions compared to the geometric mean of all available measurements for this enzyme-reaction pair. These numbers indicate that in practice, using predictions calculated with the TurNuP model may lead to similar deviations and error rates compared to performing experimental measurements. Better predictions will become possible in the future if experimental variation will be reduced through improved technologies.

Although the accuracy of TurNuP's predictions is not very different from that of experimental estimates, model accuracy can still be improved. On the one hand, we trained and validated the model with a total of only 4271 data points, which is rather small for a machine learning model with high-dimensional input vectors. Once more high-quality training data becomes available, model performance will most likely improve. On the other hand, $k_{cat}$ values can differ widely if measured under different experimental conditions such as varying pH and temperature. However, as information about the experimental conditions is mostly unavailable in databases for enzyme kinetic parameters, we were not able to include these conditions as an input to our prediction model. Manually extracting this information from research papers has the potential to further improve the accuracy of the prediction models.

TurNuP achieves superior performance compared to previous methods for predicting $k_{cat}$. Its coefficient of determination ($R^2 = 0.44$) is higher than that of Heckmann et al. ($R^2 \approx 0.34$)[14], who trained an organism-specific prediction model with very detailed and expert-crafted input features, including enzyme active site properties, metabolite concentrations, reaction fluxes, and experimental conditions. TurNuP also outperforms the most recent method for predicting $k_{cat}$, the DLKcat model[16] (Fig. 5). One reason for TurNuP's superior performance might be the use of state-of-the-art enzyme representations compared to convolutional neural networks (CNNs) and the use of representations for the whole chemical reactions instead of using only information on one of the substrates. Importantly, 91% of the enzymes in the DLKcat test set have a maximal sequence identity between 99 and 100% compared to the enzymes in the training set. It is likely that the same issue arose in the validation set used for hyperparameter optimization; such a structure of training and validation sets makes it difficult to train a model that generalizes well to enzymes not highly similar to those in the training set. Indeed, we showed that DLKcat does not produce meaningful predictions for enzymes with a maximal sequence identity lower than 40% compared to the enzymes in the training set.

In contrast, TurNuP generalizes well even to enzymes that are not highly similar to enzymes in the training set and to reactions that have not been part of the training set. To achieve these results, we used general input features: the ESM-1$b_{ESP}$ vector[17], a fine-tuned, state-of-the-art numerical representation of the enzyme, calculated from its amino acid sequence; and a reaction fingerprint that integrates structural information about all substrates and products[24], which allowed us to create input vectors of fixed length even for varying numbers of reactants.

To assess potential limitations of our methodology, we investigated the predictive capacity of TurNuP for predicting $k_{cat}$ values for membrane-bound enzymes. To this end, we analyzed prediction performance for 63 membrane proteins that are included in our test dataset, the majority of which are peripheral membrane proteins, i.e., soluble proteins that bind transiently to the surface of cell membranes.

TurNuP performs only slightly worse in predicting $k_{cat}$ for this subset of proteins compared to other proteins, achieving a coefficient of determination $R^2 = 0.36$, a mean squared error MSE = 0.68, and a Pearson correlation coefficient $r = 0.64$. This compares to a coefficient of determination $R^2 = 0.44$, a mean squared error MSE = 0.82, and a Pearson correlation coefficient $r = 0.67$ for non-membrane-associated proteins. That predictions for membrane-associated proteins combine a lower coefficient of determination with a lower mean squared error is likely related to the lower variance of $k_{cat}$ values for membrane-associated enzymes, 1.07, compared to the variance of non-membrane-associated enzymes, 1.46.

Perhaps surprisingly, $k_{cat}$ predictions that are solely based on reaction information—without information on the catalyzing enzyme—already lead to a high coefficient of determination, $R^2 = 0.38$ (Fig. 3). This result suggests that properties of the reaction have a strong influence on the turnover number achievable by natural selection on the catalyzing enzyme. Adding enzyme information additionally to the reaction information as model input had only a moderate effect on model performance, indicating that the information for predicting $k_{cat}$ that is stored in the enzyme and in the chemical reaction are strongly overlapping. It would be desirable to analyze which properties of the chemical reactions and of the enzymes are most relevant for predicting $k_{cat}$. Unfortunately, because of how we encoded the input information, it is difficult to draw conclusions about the importance of certain properties of the reactions and enzymes from analyzing the importance of the input features. For example, we could extract which input features from the ESM-1b vectors are most relevant for predicting $k_{cat}$, but we cannot easily relate entries in the ESM-1b vectors to properties of the proteins, as the ESM-1b vectors were created in a previous step by iteratively extracting information from the amino acid sequence using a deep neural network with 33 layers. However, to analyze if TurNuP learns meaningful patterns such as predicting higher $k_{cat}$ values for faster enzyme classes, we divided our dataset into different splits according to the first digit of the enzymes' EC numbers, and we compared the average measured $k_{cat}$ value with the average predicted $k_{cat}$ value for each class. Indeed, TurNuP learns to predict on average higher $k_{cat}$ values for faster EC classes and lower $k_{cat}$ values for slower enzyme classes (Supplementary Fig. 8).

It appears initially surprising that the reaction fluxes estimated with pFBA do not explain much of the variance of $k_{cat}$, while they were found to be the best predictor in the model developed by Heckmann et al.[14]. When calculating genome-scale reaction fluxes for different organisms, we obtained fluxes that were zero or close to zero for many data points. In contrast, Heckmann et al. focused on a small dataset that mostly consisted of well-studied, central reactions in *E. coli*. Those reactions typically have fluxes substantially different from zero at least in some of the simulated conditions. It appears likely that this biased construction of a small dataset in ref. [14] is responsible for the high correlation observed between reaction fluxes and $k_{cat}$ by Heckmann et al.

Computational estimates of $k_{cat}$ values are highly relevant for the functional and kinetic study of individual enzymes[42], and TurNuP can provide a first estimate of $k_{cat}$ before performing labor-intensive experiments. Another major use case of TurNuP is the prediction of $k_{cat}$ values for genome-scale metabolic models. We found that our predictions can be used successfully to improve proteome allocation predictions (Fig. 5b). In future work, $k_{cat}$ predictions with TurNuP can be combined with an existing approach for predicting Michaelis constants ($K_M$)[39]. This would facilitate full parameterizations of non-linear enzyme kinetics in genome-scale metabolic models, a powerful tool for gaining fundamental insights into cellular physiology[9,43].

## Methods

### Software and code availability

All software was coded in Python[44]. We created the enzyme representations using the deep learning library PyTorch[45]. We fitted the

gradient boosting models using the library XGBoost[34]. We used the web framework Django[46] to implement the TurNuP web server. The code used to generate the results of this paper, in the form of Jupyter notebooks, are available from https://github.com/AlexanderKroll/Kcat_prediction[47]. All datasets used to create the results of this manuscript are available from https://doi.org/10.5281/zenodo.7849347[48].

## Downloading $k_{cat}$ data

We used data from three different databases, Sabio-RK, UniProt, and BRENDA, to create a $k_{cat}$ dataset for model training and validation. We downloaded 3971 $k_{cat}$ values for wild-type enzymes together with UniProt IDs and reaction information from Sabio-RK[22]. We tried to map all metabolites involved in the reactions to unique identifiers using either a KEGG reaction ID[49], if available, or using the metabolite names and the PubChem synonym database[50]. We removed all data points for which we could not map all substrates and all products to an ID. This resulted in a dataset with 2830 data points for 289 different enzymes.

We downloaded 5664 $k_{cat}$ values for wild-type enzymes together with UniProt IDs and CHEBI reaction IDs from UniProt via the UniProt mapping service[21]. We mapped the metabolites of all reactions to unique IDs using CHEBI reaction IDs[51]. We removed data points, if we could not map all metabolites of a reaction to an ID. This resulted in a dataset with 1738 $k_{cat}$ values for 1017 different enzymes.

We downloaded 14,165 turnover numbers for wild-type enzymes with protein information and substrate names from BRENDA[20]. Most of the $k_{cat}$ values in BRENDA are not assigned with a unique reaction equation and the entered $k_{cat}$ values are known to be prone to errors[23]. To overcome these issues, we manually checked for more than half of all points if the stated $k_{cat}$ value is identical to the value from the original paper and we assigned a unique reaction equation to all manually checked data points. After removing those data points with incomplete reaction information and non unique enzyme IDs, 8267 data points were left for 3149 different enzymes.

## Data preprocessing

We merged all three $k_{cat}$ datasets from BRENDA, Sabio-RK, and Uni-Prot, which resulted in a dataset with 12,835 data points. We removed 1050 duplicated data points from this dataset. To obtain protein sequences for all enzymes, we used the UniProt mapping service[21] to map all UniProt IDs to amino acid sequences. We used the Python package Bioservices[52] to map all metabolites to InChI strings[53]. If multiple $k_{cat}$ values existed for the same enzyme-reaction combination, we took the geometric mean across these values. For the calculation of the geometric mean, we wanted to ignore those values that were likely obtained under non-optimal conditions. Thus, we excluded $k_{cat}$ values smaller than 1% compared to the maximal $k_{cat}$ value for the same enzyme-reaction combination. Calculating the geometric mean resulted in a dataset with 7496 entries.

The BRENDA, UniProt, and Sabio-RK databases contain many $k_{cat}$ values that were measured for secondary, non-natural reactions of enzymes. As we are only interested in measurements for the natural reaction of an enzyme, we excluded $k_{cat}$ values if another measurement existed for the same enzyme but for a different reaction with a $k_{cat}$ value that was more than ten times higher. To further exclude data points that were measured under non-optimal conditions or for non-natural reactions of the enzyme, we excluded data points if we could find a measurement for the same reaction or the same EC number that was more than 100 times higher. The described procedures led to the removal of 3092 data points.

We calculated reaction fingerprints and enzyme representations for all enzyme-reaction pairs (see below) and removed all 26 data points, where either the reaction fingerprint or the enzyme representation could not be calculated.

To exclude data points with possibly wrongly assigned reaction equations, we removed those 52 data points where the sum of molecular weights of substrates did not match the sum of molecular weights of the products. We removed another 55 data points because their $k_{cat}$ values are outliers (i.e., values below $10^{-2.5}$/s or higher than $10^5$/s). This resulted in a final dataset with 4271 data points.

## Splitting the dataset into training and test set

We randomly split the dataset into 80% training data and 20% test data. We made sure that the same enzyme would not occur in the training and the test set. We further split the training set into five disjoint subsets for a fivefold cross-validation (CV) to perform hyperparameter optimizations of the machine learning models. In order to achieve a model that generalizes well during CV, we created these five subsets also in such a way that the same enzyme did not occur in two different subsets.

## Calculating enzyme representations

To create the ESM-1b model[17], the Facebook AI research (FAIR) team trained a Transformer Network[30] with 33 hidden layers and a hidden layer size of 1280 using ~27 million protein amino acid sequences from the UniRef50 dataset[31]. To process a protein sequence, the type and position of every amino acid in a sequence is encoded in a 1280-dimensional numerical vector. All amino acid representations of a sequence are simultaneously applied to the ESM-1b model and updated for 33 time steps using the attention mechanism[30]. The attention mechanism allows to use all representations as an input when updating a single amino acid representation. The attention mechanism selectively chooses only relevant input when calculating an update of a representation. To train the ESM-1b model, randomly 10–15% of the amino acids in a sequence are masked. The model is then trained to predict the type of the masked amino acids. After training, a single representation for the whole model can be created by calculating the element-wise mean of all amino acid representations after they were updated for 33 times (Supplementary Fig. 3a). We used the trained ESM-1b model and the code provided on the GitHub repository of the FAIR team[17], to calculate a 1280-dimensional numerical representation for every enzyme in our dataset. As the ESM-1b model can only process amino acid sequences up to 1024 amino acids, we only used the first 1024 amino acids for those sequences that were too long.

## Calculating fine-tuned enzyme representations

As an alternative to the original ESM-1b enzyme representations[17], we employed fine-tuned enzyme representations that we created previously for the task of predicting the substrate scope of enzymes[32]. These ESM-1b$_{ESP}$ vectors were constructed using code and models provided on the following GitHub repository: https://github.com/AlexanderKroll/ESP.

We here provide a short summary of the construction. First, the original ESM-1b model was slightly modified, i.e., additionally to the input tokens for every amino acid in a protein sequence, an extra token for the whole enzyme was added. This token does not add any additional input information to the model. Instead, this token is used to store information about the whole input sequence that is salient to the downstream task. During training, all tokens, including the extra enzyme token, are first updated iteratively for 33 times, as described above for the original ESM-1b model. Next, the updated additional enzyme token was concatenated with a 1024 dimensional expert-crafted binary representation of a small molecule, the extended-connectivity fingerprint (ECFP)[54]. This concatenated vector was used as the input for a fully connected neural network that was trained to predict if the small molecule is a substrate for the given enzyme. The whole model, the ESM-1b and the fully connected layers, were trained end-to-end for the task of identifying substrates for enzymes

(Supplementary Fig. 3b). The training set consisted of 287,386 positive enzyme-substrate pairs with either phylogenetically inferred or experimental evidence extracted from the GO Annotation database[55] and with randomly sampled negative enzyme-small molecule pairs. During the training process, the model was forced to extract all relevant information for the prediction task from the amino acid sequence and store it in the updated enzyme token. This trained model was used to create the fine-tuned enzyme ESM-1b$_{ESP}$ vectors for every enzyme in our dataset by extracting the updated enzyme representation from the model.

### Calculating reaction fingerprints

To calculate difference and structural reaction fingerprints, we used functions from the RDKit[24] package. As input, this package requires reactions described in the language SMARTS[56], which describes patterns of small molecules and of chemical reactions.

Structural reaction fingerprints are created by first calculating 1638-dimensional binary molecular fingerprints (ExplicitBitVect) for all substrates and products. Then, the bitwise OR-function is separately applied to all substrate fingerprints and to all product fingerprints, which results in two 1638-dimensional binary vectors with information about the substrates and about the products, respectively. Finally, both vectors are concatenated, which results in a 3276-dimensional binary vector with structural information about the reaction. We used the RDKit function *Chem.rdChemReactions.CreateStructuralFingerprintForReaction* to calculate the fingerprints.

To calculate difference reaction fingerprints, first, a 2048-dimensional binary atom-pair fingerprint (AtompairFP) for each substrate and each product is calculated. Then, the fingerprints for all substrates and also for all products are element-wise summed. The resulting fingerprint for the products is then subtracted from the fingerprint for the substrates, which results in a 2048-dimensional reaction fingerprint with positive and negative integers. To calculate these fingerprints, we used the RDKit function *Chem.rdChemReactions.CreateDifferenceFingerprintForReaction*.

We calculated differential reaction fingerprints (DRFPs) as described in ref. 26, using the Python package drfp. Briefly, we represented all reactions in our dataset using the language SMILES. DRFPs are then calculated by identifying all substructures among all reactants up to a certain size. All substructures that are only present in either the substrates or the products are then mapped to a 2048-dimensional binary fingerprint using hash-functions.

### Calculating reaction similarity

We use the Jaccard distance to calculate the pairwise distance between two structural reaction fingerprints. The Jaccard distance is defined as the proportion of elements that disagree while considering only those entries where at least one entry is non-zero. This results in a value between 0 and 1, where lower values indicate higher similarity. To convert this distance into a similarity score, we subtracted the distance value from 1 and normalized all scores such that they would range from 0 and 1. This resulted in a similarity score where higher values indicate higher similarity between two reactions.

### Hyperparameter optimization for gradient boosting models

To perform hyperparameter optimizations for all gradient boosting models, we split the training set into five disjoint subsets with approximately equal sizes to perform fivefold cross-validations (CVs). We performed a random grid search for the hyperparameters learning rate, regularization coefficients $\alpha$ and $\lambda$, maximal tree depth, maximum delta step, number of training iterations, and minimum child weight using the Python package hyperopt[57]. Afterwards, we chose the set of hyperparameters that led to the highest mean coefficient of determination $R^2$ during CV.

### Training of additional machine learning models

To compare the performance of the gradient boosting model to additional machine learning models, we also trained a linear regression model, a random forest model, and a fully connected neural network for the same prediction task. To find the best hyperparameters for the models, we again performed 5-fold CVs on the training set. For the random forest model, the optimized hyperparameters were the number of estimators, maximal depth of trees, and minimum samples per leaf. For the linear regression model, we searched for the best $L_1$ and $L_2$ regularization coefficients. We used the Python package scikit-learn[58] for training both models. For the fully connected neural network, the optimized hyperparameters were learning rate, learning rate decay and momentum, dimension of hidden layers, batch size, $L_2$ regularization coefficient, and number of epochs. We used the deep learning library TensorFlow to train the fully connected neural networks[59].

### Comparison of $k_{cat}$ predictions between the DLKcat model and TurNuP

We used code provided on a GitHub repository by Li et al.[16] to reproduce the DLKcat model and to make predictions for their test set. We divided both, the DLKcat test set and our test set, into four different subsets according to the protein sequence identity compared to the amino acid sequences in the training sets (Fig. S2). To achieve this, we calculated for every test sequence the maximal pairwise sequence identity compared to all sequences in the training set, using the Needleman-Wunsch algorithm from the software package EMBOSS[60]. We then grouped the data points in each of the two test sets according to this maximal sequence identity: <40%; ≥40% and <80%; ≥80% and <99%; and ≥99%. For each of the two methods, we then calculated the coefficient of determination $R^2$ separately in each of the sequence identity groups. For each level of sequence identity, we compared the $R^2$ values of TurNuP (evaluated on its corresponding subset of test data points) and DLKcat (evaluated on its corresponding subset of test data points).

### Predicting protein abundances using predicted $k_{cat}$ values

Li et al.[16] developed a Bayesian pipeline to use predicted $k_{cat}$ values for enzyme-constrained genome-scale metabolic models to predict the proteome of yeast species. We used Matlab code provided on a GitHub repository by Li et al.[16] to follow the same pipeline for $k_{cat}$ values predicted with TurNuP. The predicted proteome allocations were compared to measured proteome data for four different species in 21 different cultural conditions. The measured proteome data was taken from six different publications[61-66].

### Statistical tests for model comparison

To test if the difference in model performance between the TurNuP model with enzyme and reaction information compared to the models with either only enzyme or reaction information is statistically significant, we applied a one-sided Wilcoxon signed-rank test implemented in the Python package SciPy[67]. We tested the null hypothesis that the median of the absolute errors on the test set for predictions made with TurNuP, $\bar{e}_1$, is greater or equal to the corresponding median for predictions made with a model with only reaction or only enzyme information, $\bar{e}_2$ ($H_0 : \bar{e}_1 \geq \bar{e}_2$ vs. $H_1 : \bar{e}_2 > \bar{e}_1$). We could reject $H_0$ with $p$-values of $p = 0.0049$ (DRFP) and $p = 1.2 \times 10^{-7}$ (ESM-1b$_{ESP}$)), accepting the alternative hypothesis $H_1$.

We also tested if the differences in model performance between TurNuP and the DLKcat model are statistically significant for all subsets of the test set with different enzyme sequence identity levels. We used the non-parametric one-sided Wilcoxon-Mann-Whitney test implemented in the Python package SciPy[67] to test the null hypothesis that the prediction errors for the two models are equally distributed. We

could reject the null hypothesis for all four subsets at the 5% level with $p$-values of $p = 4.3 \times 10^{-11}$ (0–40%), $p = 0.0017$ (40–80%), $p = 0.00037$ (80–99%), and $p = 0.048$ (99–100%).

## Calculating reaction fluxes

We calculated reaction fluxes for all 108 genome-scale metabolic models (GEMs) from the BiGG database[41]. We selected those six GEMs for different organisms that showed the highest correlation between calculated fluxes through parsimonious flux balance analyses (pFBA) and $k_{cat}$ values in our dataset. We selected the following six models: iECO111_1330 (*Escherichia coli*), iEK1008 (*Mycobacterium tuberculosis*), iHN637 (*Clostridium ljungdahlii*), iIT341 (*Helicobacter pylori*), iSbBS512_1146 (*Shigella boydii*), and iJN1463 (*Pseudomonas putida*).

We calculated the reaction fluxes similar to the approach by Heckmann et al.[14] for *E. coli*. For each of the six GEMs, we simulated 10 000 minimal growth sustaining environments through pFBA[38] using the Python package COBRApy[68]. Afterwards, we calculated for every reaction the mean of all non-zero fluxes among all simulations. In all of the 10 000 simulations, first a growth sustaining environment was created with a growth rate higher than 0.1 $[h^{-1}]$ and oxygen uptake was allowed with a probability of 50% for aerobic organisms. To convert the medium into a minimal media, each metabolite of the medium was removed if growth was sustained without it. If we could not obtain a non-zero flux for a reaction in all simulations, we repeated the described procedure with a flux variability analysis (FVA)[69] instead of a pFBA. If we could not obtain a non-zero flux for a reaction either via pFBA or via FVA, we replaced the reaction flux with the mean of all non-zero fluxes. Python code for calculating the fluxes is available on the following GitHub repository: https://github.com/Nina181/kcat_flux_relationship.

## Mapping data points to BiGG reaction IDs

We created a list with reactions from six different metabolic genome-scale models from the BiGG database[41] (iECO111_1330, iEK1008, iHN637, iIT341, iSbBS512_1146, iJN1463). To create this list, we downloaded a JSON-files for each model and we extracted all substrate names and IDs (MetaNetX or KEGG), product names and IDs, and BiGG reaction IDs. We discarded all reactions with an incomplete list of substrate or product IDs. If only a MetaNetX ID and no KEGG ID was available for a metabolite, we downloaded an InChI string[53] for the metabolite using the MetaNetX database[70]. Next, we calculated structural reaction fingerprints for all extracted BiGG reactions using the KEGG IDs and InChI strings of the substrates and products (for details see above, "Calculating reaction fingerprints").

To map data points from our dataset to BiGG reaction IDs, we calculated a pairwise similarity score between all reactions in our dataset and all reactions from the 6 extracted BiGG models. To calculate the similarity score, we used the Python function *TanimotoSimilarity* from the RDKit package DataStructs[24] with structural reaction fingerprints as the input. This resulted in a similarity score between 0 (no similarity) and 1 (very high similarity) for all pairs of reactions. We mapped every data point in our dataset to the BiGG reaction with the highest similarity score.

## Calculating Michaelis constants

To calculate the Michaelis constants $K_M$ for all enzyme-catalyzed reactions in our dataset, we created a list with all enzyme-substrate pairs. We used the BRENDA database[20] to map enzyme-substrate pairs to $K_M$ values via the enzymes' amino acid sequences and via a molecular fingerprint of the substrate, called ECFP vector[54]. We were able to map a $K_M$ value to ~7% of 8984 enzyme-substrate pairs.

If we could not find a value for an enzyme-substrate pair in the BRENDA database, we predicted $K_M$ using a machine learning model[39]. The $K_M$ prediction model uses a graph neural network (GNN)[71,72] to create a 50-dimensional task-specific fingerprint of the substrate.

These fingerprints are used together with a 1900-dimensioanl enzyme representation, called UniRep vector[29], as the input for a gradient boosted decision tree model[34] to predict the $K_M$ value for an enzyme-substrate pair. For reactions with multiple substrates, we took the geometric mean of $K_M$ values to create a single $K_M$ value for every data point.

## Calculating the codon adaptation index

The codon adaptation index (CAI) for *E. coli* was calculated considering ribosomal protein genes as the highly expressed genes[40]. The sequences of ribosomal protein genes were retrieved from genome annotation of *E. coli* (NC_000913.3 from RefSeq[73]).

## Computational resources

To perform hyperparameter optimization for all gradient boosting models, we used the High Performance Computing Cluster at the University of Düsseldorf (Germany). Each hyperparameter optimization was executed for 48 hours on an Nvidia Quadro RTX 8000, which resulted in testing ~5000 different hyperparameter settings for each model.

## Reporting summary

Further information on research design is available in the Nature Portfolio Reporting Summary linked to this article.

## Data availability

We used the BRENDA[20], UniProt[21], and Sabio-RK[22] databases to create a $k_{cat}$ dataset. All data downloaded and generated in this study and all processed data used to produce the results of this study have been deposited in a Zenodo repository available at https://doi.org/10.5281/zenodo.7849347[48]. Source data for all figures are provided with this paper. Source data are provided with this paper.

## Code availability

The Python code used to generate all results is publicly available only at https://github.com/AlexanderKroll/kcat_prediction[47].

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

## Acknowledgements
We thank David Heckmann for helpful discussions. Computational support and infrastructure was provided by the "Centre for Information and Media Technology" (ZIM) at the University of Düsseldorf (Germany). We acknowledge financial support to MJL by the Deutsche Forschungsgemeinschaft (DFG, German Research Foundation) through CRC 1310, and, under Germany's Excellence Strategy, through grant EXC 2048/1 (Project ID: 390686111).

## Author contributions
A.K. designed the dataset and models and performed all other analyses. N.A.L. implemented the calculation of the reaction fluxes. X.P.H. calculated the codon adaptation index (CAI) for genes from *E. coli*. Y.R. calculated the differential reaction fingerprints (DRFPs) and analyzed their predictive power. M.J.L. conceived of and supervised the study, and acquired funding. A.K. wrote the initial manuscript, which was edited by A.K. and M.J.L.

## Funding

## Competing interests
The authors declare no competing interests.
