## [Peer Review File · Nature Communications]

Turnover number predictions for kinetically uncharacterized enzymes using machine and deep learningREVIEWER COMMENTS

Reviewer #1 (Remarks to the Author):

In this work, the authors present TurNuP, a machine learning-based approach for kcat prediction using reaction chemical fingerprints and an unsupervised deep learning representation of enzyme sequence data as features. They used typical kinetics databases as training data (BRENDA, SABIO-RK, UniProt), excluding mutant data. They compare their model to a previous tool, DLKcat. They test the functional performance of their predicted kcats in proteome allocation models.

kcats are key parameters in kinetic and constraint-based metabolic modeling efforts, which have seen a resurgence in recent years especially around questions of proteome allocation. Accurate estimation of kcats across diverse organisms is of great interest to the community.

The approach utilized by the authors seems quite powerful in general for predicting kcats for as wide a range of enzymes as possible. The data combination and filtration steps seem mostly logical, although I have a few questions on the details here. I tested out the webserver, and it worked quickly for a single query. The Python code available on Github is appreciated and includes workflows with results embedded, greatly improving the reproducibility of the work.

In total, this looks like another great addition to the growing body of work on kcat estimation. I have a number of fairly moderate concerns about the details of the methods and the presentation of results, which if addressed should help improve the impact of this work on the community. Most serious among these, to me, is the fact that the authors seem to compare the absolute R^2 performance across methods without consideration for the fact that different test data sets are used to compute these scores. Details on these and other comments are below:

Major Comments

- Processing of training data: The authors are admirably focused on setting up a valid training set, avoiding having an identical sequence in both the training and testing. For example, if the dataset contains multiple copies of a single E.C. for different conditions, organisms etc, only one value should be included, which if I understand correctly, the authors handle. They state "We split the dataset into 80% training data and 20% test data in such a way that enzymes with the same amino acid sequence would not occur both in the training and in the test set." Does this guarantee that the model isn't trained on one version of an enzyme from a certain species and used to predict a very slightly different version in the test set in a different species? The authors state their purpose is this (they mention wanted to avoid predicting 'close homologs' artificially well), but the fact that they use completely identical sequences as the criteria makes me wonder, rather than sequence similarity greater than X%. I would love to take a look at the training and testing datasets, but I couldn't find them in either a Supplementary Dataset or on the Github. Also in Figure 3 – it is unexpected to me that the performance is higher on the test set than the majority of training sets – are they sure there is no overfitting due to memorized data?

- Rationale for reaction fingerprints: For the reactant fingerprints (which admittedly were not used in the final model), why is the OR a good operator to use to combine substrates? What are the features being represented in a binary fashion and why would presence of any of these features be a good way to represent combined reactant properties? I don't understand the theoretical basis here. I understand that the variable number of reactants presents a hurdle in encoding reactant features, but it just seems like the sum used in the reaction fingerprints, or an empty feature set of a certain fixed size to allow up to 3 substrates, would be more promising approach. It just seems a shame that both reactant and reaction features couldn't be both included, as one would think that for example, a transaminase reaction operating on a small substrate would have a faster possible kcat than the same reaction operating on a large substrate, due to diffusion differences if nothing else.

- Details of transformer model methods: I am not clear on how the fine tuning works to generate ESM-1bESP vectors – what is the metric to fine tune and why would it improve performance? Line 452 is completely insufficient if that is the only description of methods provided.

- Analysis of learned features: I was expected or hoping to see more analysis of what the model actually learns as important to kcats. Do things make sense, i.e. slow and fast classes of reactions? What are the effects of primary vs cofactor substrates etc? Presumably the sequence

info is difficult to analysis but it would still be nice to see some kinds of trends. Which features are most useful in the predictor and what do they mean? Do they correspond to particular folds, overall size/hydrophobicity of the protein or what? How much does the target data represent the space of predicted data? i.e. sequence similarity, substrate similarity etc? Are the greatly extrapolating or mostly interpolating? – they do mention that it can extrapolate – what is the basis for this learning? Something like membrane proteins are unlikely to be trainable in the same model, or are they? Can these be examined to see how well they work?

- Sensitivity/error on predictions: More discussion on where it fails or succeeds would be nice, beyond simple sequence identity. Providing sensitivities or expected error with predictions would be a big improvement. As a quick test, I used the server to estimate the well-known E. coli enzyme pfkA, which presumably is part of the training set (or other PFK enzymes), using the protein sequence from UniProt and KEGG IDs for the reactants. The prediction was $\sim 6 \text{ s}^{-1}$, while the actual kcat is above 100 in both in vitro (PMID: 1429704) and in vivo estimates (PMID: 26951675), so the prediction was not particularly impressive for this single case. Perhaps the authors could comment on this – are well-known enzymes not expected to be predicted accurately by the tool? Is it possible to know when a prediction is less likely to be accurate?

- Use of statistics: The use of statistics throughout the manuscript seems unnecessary and problematic. For example, Lines 158-161 – “However, a p-value below 0.5 indicates that rejecting the null hypothesis – that ESM-1bESP vectors do not lead to superior results – likely improves the model.” The interpretation here is rather suspect to put it mildly, as $p < 0.5$ does not mean that your null hypothesis is more likely than not to be rejected, it just means that you cannot reject your null hypothesis. It would be more straightforward to just report the absolute values of different performance metrics rather than to worry about assigning a p-value to the difference. This goes for Line 174 and other uses of p-values to compare model performance – it just feels unnecessary and conceptually problematic.

- Comparisons of TurNuP to other methods: DLKcat was mentioned and compared quantitatively but a more high level description of the differences in how the methods work would be welcome. More seriously, the quantitative comparisons between methods are problematic throughout the manuscripts because the test datasets are different. For example, Lines 215-220 – these are completely meaningless comparisons because the test set is different. If the authors want to compare absolute R^2 values across methods, they need at the very least to test the methods on the same data, of course ensuring that the data does not lie in the training set as well. Similarly, the DLKcat comparison again seems to use a different test set if I understand correctly.

- Global underprediction of high kcats: The distribution in figure 4 looks like their model underpredicts higher kcats substantially – a 1,1 line should be drawn on this figure as well as a correlation line. Is there any explanation for what seems to be a systemic bias in the model?

Minor Comments

- Alternate explored features: The section exploring flux, CAI, and Km as alternative features is weak and probably deserved to be relegated to the discussion. The methods for estimating these parameters are not well tested and subject to their own problems. For example, the use of Kms as a single data point for all reactants during prediction seems greatly flawed and in serious need of validation studies. Error in estimation of these parameters could easily be the sole cause of their failure to aid model predictions, regardless of whether the parameters do have a relationship with kcats. Including the existing results is not problematic per se, but I am worried that readers would confuse a lack of success of these estimated features as implemented to indicate that these features do not relate to kcat. For example, I do not believe they have sufficient evidence to make the statement starting on line 360 in the Discussion.

- Another prominent approach to kcat estimation is through in vivo kcat estimation using proteomics data. The authors cite the originating paper by Davidi et al but only in another context. Some discussion of this alternate approach and differences in achieved accuracy would be appropriate.

- I didn't understand the description in the methods of how they compared DLKcat to TurNuP, specifically these lines from 482-486. Could the authors expand on this explanation or provide a figure on the methods?

- How are reaction conditions (temperature) that could systematically affect kcat values handled? What is the intended temperature of the predictions? 25C?

- No Supplementary dataset? Including some of their data in tables would improve the use of their work by the community, for example flux predictions, CAI values, etc.

- Supplementary plots should show best fit line and correlation if the intention is to show a relationship or lack thereof between variables.
- Figure S1 on normal distributions – why not show a Q-Q plot that is actually intended to show normality, comparing to a normal distribution with the same mean and variance as the observed distribution? Not that this particularly matters – the log transformation is certainly a good thing for model training.
- Their website (<https://turnup.cs.hhu.de/>) has a typo in its title ('Turnover Number Prediction - TNP') and when you run a job, the message has a typo "Your result is currently being calculated."
- Around line 68, some Figure formatting ("Figure ??") (recurring issue, maybe with the Supplementary Figures)

Reviewer #2 (Remarks to the Author):

Kroll et al train a deep-learning-based network to predict kcat values from a representation of the substrate or the protein or their combination. Accurate kcat prediction would be valuable for metagenomic screening and enzyme engineering studies, but I see significant problems with the current submission.

Major comments:

- * It is not clear to me how well the data were separated into training and test sets and it seems that the criteria were rather loose. From page 4, I'm led to believe that only precisely the same amino acid sequence would be excluded, but it is clear that if a method were trained on data on a homolog then it has a very good prior for estimating the activity of the query. I'm not sure that there's a good way out of this conundrum since generating a truly nonredundant set (<35% sequence identity) is likely to lead to a very small training set.
- * On pg. 5, the authors demonstrate that training on the substrate alone leads to a high success rate in prediction. They don't draw conclusions from this observation (in general, the manuscript is very narrowly focused on the statistical analysis and provides little in biochemical insight). This may imply that, similar to the above point, the method is in effect learning to predict reactivity based on homology, which is a trivial outcome of the training process.
- * The data presented in Fig. 4 correlates predicted versus measured kcat values shows that the correlation is strong in log scale but that there is substantial noise within an order of magnitude. This is perhaps to be expected but the authors should acknowledge this, because in many enzymology studies, a much higher accuracy would be needed. Second, it may help their case if they showed that measurements of the same protein in two different studies (where such data exist) are also noisy. I believe this to be the case but I don't know what magnitude of error to expect.
- * As a very general comment, the paper is not written well enough in my opinion. There are many typos throughout (even the first line of the Introduction), and it is very lengthy compared to the amount of information and insight that is presented.
- * On line 160, the authors seem to acknowledge that they chose their models based on the resulting p-value. This is known as p-value hacking, and if they employed this strategy, they should make it explicit that they used much more stringent p-values than are otherwise common.

Reviewer #3 (Remarks to the Author):

The topic is well-grounded as an incremental work from Nature Catalysis [16]. The deep transformer learning methodology looks appropriate. The results are minimally enough. The deliverables are very nice, including website, source code, and clearly reproducible procedure details in the writing. Taking into the above, I guess the submission could be revised. My comments are as follows:

Major Comments:

The only competing method to be compared is DLKcat which is published on Nature Catalysis [16] [16] Li, F. et al. Deep learning-based kcat prediction enables improved enzyme-constrained model 605 reconstruction. Nat. Catal., 1–11 (2022)

I was wondering if other possible approaches should be compared; for instance, other machine learning and deep learning approaches? Baseline estimations are needed on top of the comparisons to [16].

The authors heavily focused on performance gains (i.e. minimizing errors). Assuming the proposed method is fine, would there be new / novel insight into the problem domain (i.e. enzymatic turnovers) ?

Deep learning is known to be slow. Running time analysis could be depicted with the hardware setup details.

Deep learning methodology reviewers may argue the neural network architecture should be visualized and examined for further details with ablation analysis. I am flexible with it though.

Statistical significance testing could be conducted; for instance, p-value calculations across multiple runs of cross-validations.

Minor Comments:

I can see there are formatting errors " (Figure ??). " on page 14/30.

An example sequence should be given in the webpage for user-friendly usage.

I cannot find the complete source code download on the webpage, but on the GitHub link provided in the Methods section. Standalone executables on Linux and Windows could also be provided.

The webpage looks very simple and does not contain enough documentations and tutorials.

Reviewer #1 (Remarks to the Author):

In this work, the authors present TurNuP, a machine learning-based approach for k_{cat} prediction using reaction chemical fingerprints and an unsupervised deep learning representation of enzyme sequence data as features. They used typical kinetics databases as training data (BRENDA, SABIO-RK, UniProt), excluding mutant data. They compare their model to a previous tool, DLKcat. They test the functional performance of their predicted k_{cat} s in proteome allocation models.

k_{cat} s are key parameters in kinetic and constraint-based metabolic modeling efforts, which have seen a resurgence in recent years especially around questions of proteome allocation. Accurate estimation of k_{cat} s across diverse organisms is of great interest to the community.

The approach utilized by the authors seems quite powerful in general for predicting k_{cat} s for as wide a range of enzymes as possible. The data combination and filtration steps seem mostly logical, although I have a few questions on the details here. I tested out the webserver, and it worked quickly for a single query. The Python code available on Github is appreciated and includes workflows with results embedded, greatly improving the reproducibility of the work.

In total, this looks like another great addition to the growing body of work on k_{cat} estimation. I have a number of fairly moderate concerns about the details of the methods and the presentation of results, which if addressed should help improve the impact of this work on the community. Most serious among these, to me, is the fact that the authors seem to compare the absolute R^2 performance across methods without consideration for the fact that different test data sets are used to compute these scores. Details on these and other comments are below:

Response: We thank the reviewer for the positive assessment and for the extensive and thoughtful comments provided. We greatly appreciate the time and effort the reviewer took to provide such high-quality feedback, and we found the comments to be extremely helpful, especially in identifying areas that required further explanation. We believe that the feedback has helped us to significantly improve our manuscript.

Major Comments

1. **(a) Processing of training data:** The authors are admirably focused on setting up a valid training set, avoiding having an identical sequence in both the training and testing. For example, if the dataset contains multiple copies of a single E.C. for different conditions, organisms etc, only one value should be included, which if I understand correctly, the authors handle. They state “We split the dataset into 80% training data and 20% test data in such a way that enzymes with the same amino acid sequence would not occur both in

the training and in the test set.” Does this guarantee that the model isn’t trained on one version of an enzyme from a certain species and used to predict a very slightly different version in the test set in a different species? The authors state their purpose is this (they mention wanted to avoid predicting ‘close homologs’ artificially well), but the fact that they use completely identical sequences as the criteria makes me wonder, rather than sequence similarity greater than X%. **(b)** I would love to take a look at the training and testing datasets, but I couldn’t find them in either a Supplementary Dataset or on the Github. **(c)** Also in Figure 3 – it is unexpected to me that the performance is higher on the test set than the majority of training sets – are they sure there is no overfitting due to memorized data?

Response (a): We indeed split the dataset into a training and a test set such that no identical protein is in both sets. As pointed out by the reviewer, this procedure does not guarantee that no close homologs are in the training and the test sets. However, while we did not use a sequence similarity threshold to create the test set, calculating enzyme sequence identities for all proteins in the test set shows that only very few test proteins (2.5%) are highly similar to training proteins (Figure 5a). Moreover, to quantify the severity of the potential problem raised by the reviewer, we analyzed test performance for different levels of sequence identity between test proteins and proteins in the training set (Figure 5a). This analysis shows – at least approximately – how test performance would vary for test sets created with different sequence identity thresholds. Nevertheless, we agree that we should make it clear that close homologs are in some cases present in the test set.

Action (a): We now explicitly state that our splitting procedure does not exclude close homologs from the test set. Moreover, we explain in more detail why this is not a substantial problem: because (i) we show that the test set does not contain many proteins that are highly similar to proteins in the test set, and (ii) we separately investigate model performance for those test proteins without close homologs in the training set. These changes are found in l. 76ff:

This splitting procedure does not prevent the inclusion of enzymes in the test set with close homologs in the training set. However, we found that only very few enzymes in the test set are highly similar to enzymes in the training set, i.e., only 70 out of 850 enzymes in the test set show $\geq 90\%$ sequence identity to enzymes in the training set, and only 26 enzymes show $\geq 98\%$ identity. Moreover, to evaluate model performance for different levels of enzyme similarities (see below), we divided the test set further into subsets with different levels of maximal sequence identities between training and test enzymes (Figure S2).

Response (b): We apologize for making it difficult to find our training and test set. For the previous version of the manuscript, we had uploaded all datasets to a cloud, because we could not store all data on GitHub. The link is provided on the GitHub repository. Training and test datasets are stored in the subfolder “\data\kcat_data\splits”.

Action (b): To facilitate an easier access to the data, we now uploaded all data to Zenodo (<https://doi.org/10.5281/zenodo.775531>), and we provide a link to the data repository in our GitHub repository and in the manuscript (l. 522f.):

All datasets used to create the results of this manuscript are available from <https://doi.org/10.5281/zenodo.775531>.

Response (c): The model does not perform better on the test set than on the training set. In Figure 3a, the boxplots display the results of our 5-fold cross validation (CV). To calculate the results for Figure 3a, the model is validated on data that is originally part of our “training data set” and has been used for hyperparameter optimization, but it has not been used to train the model during this round of CV. The boxplots thus display results on data that has not been used for model training. It may still seem surprising that model performance on the fully independent test set is better than the results during CV. One possible reason is that for validating the model on the test set, more training data (~700 additional data points) is used than during CV.

Action (c): We added a short paragraph to the manuscript explaining why model performance during CV can be lower compared to model performance on the total test set (l. 244ff.):

It is noteworthy that the accuracies on the test set are partly higher than the accuracies achieved during CV (Figure 3a). To calculate the results of the CV, the model is validated on data that is originally part of our training data set and has been used for hyperparameter optimization, but it has not been used to train the model during this round of CV. The improved performance on the test set may result from the fact that before validation on the test set, models are trained with approximately 700 more samples than before each cross-validation; the number of training samples likely has a substantial influence on model performance.

2. Rationale for reaction fingerprints: For the reactant fingerprints (which admittedly were not used in the final model), why is the OR a good operator to use to combine substrates? What are the features being represented in a binary fashion and why would presence of any of these features be a good way to represent combined reactant properties? I don't understand the theoretical basis here. I understand that the variable number of reactants presents a hurdle in encoding reactant features, but it just seems like the sum used in the reaction fingerprints, or an empty feature set of a certain fixed size to allow up to 3 substrates, would be more promising approach. It just seems a shame that both reactant and reaction features couldn't be both included, as one would think that for example, a transaminase reaction operating on a small substrate would have a faster possible k_{cat} than the same reaction operating on a large substrate, due to diffusion differences if nothing else.

Response: We want to emphasize that our methodology did not involve the development of the structural reaction fingerprints themselves, but rather the application of pre-existing rules for their calculation. The OR operator – as introduced by Landrum et al. in RDKit, Ref. 24 – allows to gather the information about the substrates/products in single vectors. Binary entries of the reactant fingerprints can, for example, encode the information if a certain substructure (such as a ring) is present in a molecule. Thus, the OR operator summarizes which properties are present among the substrates and products. We completely agree that substantial information is lost

during this process – in particular, properties from different substrates/products are no longer distinguishable. We also agree that separately encoding information for every reactant would be more desirable in principle. However, the input dimension is already very large with these condensed reaction fingerprints (1280 (ESM-1b) + 2048 (reaction FP) = 3328 dimensions total) compared to the limited number of data points in our training set. If we would allocate, for example, a 1024-dimensional fingerprint for each reactant in the input feature space for a reaction that can contain up to 3 substrates and products, this would result in an input vector of dimension $1280+6*1024=7424$, which would appear too large given the number of available training data points.

Moreover, an advantage of the used reaction fingerprints is that the output is invariant to the order of the substrates and the products. If encoding each reactant separately, the resulting input feature will not be order invariant and the machine learning model has to learn that the order in which the features for the substrates/products are presented is not relevant.

Prompted by the Reviewer's comment, we have now explored a third type of reaction fingerprint. These differential reaction fingerprints (DRFP) are based on a numerical representation of structural features that are only present in either the substrates or the products. Thus, they explicitly reflect features of both substrates and products, focusing on which features actually change in the reaction. We found that this additional fingerprint type indeed leads to improved performance, with the new results fully supporting our previous results and conclusions.

Action: We now explain the motivation for using the chosen reaction fingerprints more clearly. We also clearly state that important information can be lost when creating these fingerprints, and we discuss why encoding each reactant separately is not practical in our case (ll. 110ff.):

Creating the three reaction fingerprints described above can result in information loss. For example, if multiple substrates or products are present, it is not possible to distinguish between properties of different reactants. However, the fingerprints allow to store information about reactions in a fixed-length vector even for varying numbers of reactants. Storing every reactant in a separate fingerprint would result in much larger input vectors for reactions with multiple substrates and products. Moreover, the resulting vectors would not be invariant to the order of the substrates and products and would vary in length for varying numbers of reactants.

Moreover, we introduce a third type of reaction fingerprint, DRFP, and show that it leads to slightly better predictions than the two alternatives employed previously (ll. 105 ff., ll. 135 ff., Fig. 3, and throughout the manuscript in terms of updated results):

The differential reaction fingerprints (DRFPs) are calculated in a similar fashion compared to the two fingerprints described above. However, instead of first calculating fingerprints for each reactant and then combining these into a single reaction vector, a fingerprint for the whole reaction is directly created. To achieve this, substructures of all substrates and products are identified. All substructures that are only present either in the substrates or the products are then mapped to a 2048-dimensional binary fingerprint using hash-functions.

[...]

As the DRFPs led to improved performance on the test set and during CV (Figure 3), we chose DRFPs to represent the catalyzed chemical reactions in the further analyses. To test if the better performance of the DRFPs is statistically significant, we used a two-sided Wilcoxon signed-rank test that compared the absolute errors of the models on the test set, resulting in $p = 0.064$ (DRFP vs. difference fingerprints) and $p = 2.61 \times 10^{-4}$ (DRFP vs. structural fingerprint). Thus, while the difference in absolute errors between DRFPs and difference reaction fingerprints is not statistically significant at the commonly used 5% level, a value close to 0.05 indicates that the DRFPs indeed likely lead to improved model performance.

3. Details of transformer model methods: I am not clear on how the fine tuning works to generate ESM-1b_{ESP} vectors – what is the metric to fine tune and why would it improve performance? Line 452 is completely insufficient if that is the only description of methods provided.

Response: We apologize for providing too little information on how these enzyme representations were created.

Action: We created a new subsection in the Methods section, “Calculating fine-tuned enzyme representations”, where we describe how the fine-tuned enzyme representations were created (ll. 590 ff.):

As an alternative to the original ESM-1b enzyme representations, we employed fine-tuned enzyme representations that we created previously for the task of predicting the substrate scope of enzymes. These ESM-1b_{ESP} vectors were constructed using code and models provided on the following GitHub repository: <https://github.com/AlexanderKroll/ESP>.

We here provide a short summary of the construction. First, the original ESM-1b model was slightly modified, i.e., additionally to the input tokens for every amino acid in a protein sequence, an extra token for the whole enzyme was added. This token does not add any additional input information to the model. Instead, this token is used to store information about the whole input sequence that is salient to the downstream task. During training, all tokens, including the extra enzyme token, are first updated iteratively for 33 times, as described above for the original ESM-1b model. Next, the updated additional enzyme token was concatenated with a 1024 dimensional expert-crafted binary representation of a small molecule, the extended-connectivity fingerprint (ECFP). This concatenated vector was used as the input for a fully connected neural network that was trained to predict if the small molecule is a substrate for the given enzyme. The whole model, the ESM-1b and the fully connected layers, were trained end-to-end for the task of identifying substrates for enzymes (Figure S3b). The training set consisted of 287386 positive enzyme-substrate pairs with either phylogenetically inferred or experimental evidence extracted from the GO Annotation database and with randomly sampled negative enzyme-small molecule pairs; note that phylogenetic evidence was only

used to construct the ESM-1b model, but was not used to train or evaluate the final TurNuP model. During the training process, the model was forced to extract all relevant information for the prediction task from the amino acid sequence and store it in the updated enzyme token. This trained model was used to create the fine-tuned enzyme ESM-1b_{ESP} vectors for every enzyme in our dataset by extracting the updated enzyme representation from the model.

Additionally, we created a Figure displaying the architecture of the ESM-1b and the fine-tuned ESM-1b model (Supplementary Figure S3).

4. **(a)** Analysis of learned features: I was expected or hoping to see more analysis of what the model actually learns as important to k_{cat} s. Do things make sense, i.e. slow and fast classes of reactions? What are the effects of primary vs cofactor substrates etc? Presumably the sequence info is difficult to analysis but it would still be nice to see some kinds of trends. Which features are most useful in the predictor and what do they mean? Do they correspond to particular folds, overall size/hydrophobicity of the protein or what? **(b)** How much does the target data represent the space of predicted data? i.e. sequence similarity, substrate similarity etc? Are the greatly extrapolating or mostly interpolating? – they do mention that it can extrapolate – what is the basis for this learning? **(c)** Something like membrane proteins are unlikely to be trainable in the same model, or are they? Can these be examined to see how well they work?

Response (a): We agree that it would indeed be extremely interesting to understand what the model actually learns. Unfortunately, because of the not easily interpretable encoding of the input information, it is difficult to draw conclusions about the properties of the reactions and enzymes from feature importances. For example, information extracted via deep learning from the protein amino acid sequences is indeed very difficult to analyze. We could extract which input features from the ESM-1b (or ESM-1b_{ESP}) vectors are most relevant for predicting k_{cat} , but we cannot easily relate these features to properties of the proteins. This is because the ESM-1b vectors were created in a previous step by iteratively extracting information from the amino acid sequence using a deep neural network with 33 layers. However, we agree that it is interesting to analyze if the model is learning patterns that make sense, such as predicting higher k_{cat} values for faster enzyme classes.

Action (a): We added a paragraph in the discussion section explaining why it is difficult to connect k_{cat} predictions to properties of the proteins and reactions (ll. 488 ff.):

It would be desirable to analyze which properties of the chemical reactions and of the enzymes are most relevant for predicting k_{cat} . Unfortunately, because of how we encoded the input information, it is difficult to draw conclusions about the importance of certain properties of the reactions and enzymes from analyzing the importance of the input features. For example, we could extract which input features from the ESM-1b vectors are most relevant for predicting k_{cat} , but we cannot easily relate entries in the ESM-1b vectors to properties of the proteins, as

the ESM-1b vectors were created in a previous step by iteratively extracting information from the amino acid sequence using a deep neural network with 33 layers.

We also added a short analysis showing that TurNuP predicts on average higher k_{cat} values for faster enzyme classes and lower k_{cat} values for slower EC classes (ll. 473 ff.; Supplementary Figure S8):

However, to analyze if TurNuP learns meaningful patterns such as predicting higher k_{cat} values for faster enzyme classes, we divided our dataset into different splits according to the first digit of the enzymes' EC numbers, and we compared the average measured k_{cat} value with the average predicted k_{cat} value for each class. Indeed, TurNuP learns to predict on average higher k_{cat} values for faster EC classes and lower k_{cat} values for slower enzyme classes (Figure S8).

Response (b): In the Results section “TurNuP provides meaningful predictions even if no close homologs with known k_{cat} exist”, we show that the model provides on average good predictions for amino acid sequences that are not similar to sequences in the training set, i.e., it extrapolates well to these unseen enzymes that have no close homologs in the training set.

We thank the reviewer for suggesting to perform a similar analysis for the similarity of reactions in the training set compared to the reactions in the test set to analyze TurNuP's capabilities to extrapolate to previously unseen reactions.

Action (b): We added a subsection to the Results section analyzing TurNuP's capabilities to extrapolate to previously unseen reactions (“Good predictions even for unseen reactions”, ll. 277ff.):

In the previous subsection, we showed that model performance is highest for enzymes that are similar to proteins in the training set. Similarly, it appears likely that the model performs better when making predictions for chemical reactions that are also in the training set. To test this hypothesis, we divided the test set into data points with reactions that occurred in the training set ($N = 354$) and those with reactions that did not occur in the training set ($N = 496$).

Unsurprisingly, TurNuP performs better for those data points with reactions that occurred in the training set, achieving a coefficient of determination $R^2 = 0.57$, a mean squared error $\text{MSE} = 0.51$, and a Pearson correlation coefficient $r = 0.78$ (Figure 6). However, for those test data points with reactions that TurNuP has not seen during training, model performance is still good, resulting in a coefficient of determination $R^2 = 0.35$, mean squared error $\text{MSE} = 1.02$, and Pearson correlation coefficient $r = 0.60$.

For those test data points with reactions not present in the training set, we wondered if a high similarity of the reaction compared to at least one reaction in the training set leads to improved predictions, analogous to what we observed for enzymes with higher sequence identities. For each reaction not present in the

training set, we calculated a maximal pairwise similarity score compared to all reactions in the training set based on their structural reaction fingerprints. We indeed found that prediction performance is higher for those data points with reactions more related to training reactions (Figure 6). Even for reactions that are not highly similar to training reactions, i.e., reactions with a similarity score between 0.4 and 0.8, prediction accuracy is still moderate with a coefficient of determination $R^2 = 0.20$, a mean squared error $MSE = 1.30$, and a Pearson correlation coefficient $r = 0.47$. Only for those 17 test data points with reactions that share almost no similarity compared to training reactions, model performance is low, achieving a coefficient of determination $R^2 = 0.06$, a mean squared error $MSE = 1.26$, and a Pearson correlation coefficient $r = 0.29$. We conclude that TurNuP can provide useful predictions for most unseen reactions even if these are not highly similar to previously seen reactions.

Response (c): Membrane proteins are part of our training and test set. To the best of our knowledge, very few transmembrane proteins have enzymatic functions. Accordingly, most membrane-associated enzymes in our dataset are peripheral membrane proteins, i.e., soluble proteins that bind transiently to the surface of cell membranes.

Action (c): We added a paragraph in the Discussion section, where we separately analyze model performance for membrane-associated proteins compared to non-membrane-associated proteins (ll. 471ff.):

To assess potential limitations of our methodology, we investigated the predictive capacity of TurNuP for predicting k_{cat} values for membrane-bound enzymes. To this end, we analyzed prediction performance for 63 membrane proteins that are included in our test dataset, the majority of which are peripheral membrane proteins, i.e., soluble proteins that bind transiently to the surface of cell membranes. TurNuP performs only slightly worse in predicting k_{cat} for this subset of proteins compared to other proteins, achieving a coefficient of determination $R^2 = 0.36$, a mean squared error $MSE = 0.68$, and a Pearson correlation coefficient $r = 0.64$. This compares to a coefficient of determination $R^2 = 0.44$, a mean squared error $MSE = 0.82$, and a Pearson correlation coefficient $r = 0.67$ for non-membrane-associated proteins. That predictions for membrane-associated proteins combine a lower coefficient of determination with a lower mean squared error is likely related to the lower variance of k_{cat} values for membrane-associated enzymes, 1.07, compared to the variance of non-membrane-associated enzymes, 1.46.

5. (a) Sensitivity/error on predictions: More discussion on where it fails or succeeds would be nice, beyond simple sequence identity. Providing sensitivities or expected error with predictions would be a big improvement. As a quick test, I used the server to estimate the

well known *E. coli* enzyme pfkA, which presumably is part of the training set (or other PFK enzymes), using the protein sequence from UniProt and KEGG IDs for the reactants. The prediction was $\sim 6 \text{ s}^{-1}$, while the actual k_{cat} is above 100 in both in vitro (PMID: 1429704) and in vivo estimates (PMID: 26951675), so the prediction was not particularly impressive for this single case. Perhaps the authors could comment on this – are well known enzymes not expected to be predicted accurately by the tool? **(b)** Is it possible to know when a prediction is less likely to be accurate?

Response (a): Unfortunately, the *E. coli* enzyme pfkA is neither part of our training nor test set, and the most similar enzyme in our dataset has a sequence identity of $\sim 26\%$. However, we found that the same reaction (ATP + D-Fructose-6-Phosphat \rightarrow ADP + D-Fructose-1,6-bisphosphat) is part of our training set, catalyzed by an enzyme from the organism *Solanum chacoense* (Chaco potato) with the UniProt ID Q3S2I3. This enzyme has an experimentally measured k_{cat} value of 3.78/sec. We believe this is the reason that TurNuP predicted such a low value for the *E. coli* enzyme pfkA.

Action (a): None

Response (b): We have shown that for enzymes with higher sequence identity compared to enzymes in the training set, model performance is higher. In the revised manuscript, we also investigate model performance for reactions that have been part of the training set compared to reactions that have not been present during training, concluding that model performance improves if the reaction or a similar reaction was included during training. Thus, the enzyme sequence identity as well as a reaction similarity score can be used to estimate the accuracy of a prediction.

Action (b): To allow users of our prediction tool to estimate how accurate certain predictions are likely to be, the web server now outputs the maximal enzyme sequence identity compared to all training sequences as well as a maximal reaction similarity score compared to all training reactions (ll. 408ff.):

Additionally to a k_{cat} prediction, the web server outputs the maximal enzyme sequence identity compared to all training enzymes and a maximal reaction similarity score compared to all training reactions. As discussed above, higher scores indicate higher prediction performance, and thus, these scores can be used to estimate the accuracy of the k_{cat} prediction.

6. Use of statistics: The use of statistics throughout the manuscript seems unnecessary and problematic. For example, Lines 158-161 – “However, a p-value below 0.5 indicates that rejecting the null hypothesis – that ESM-1b_{ESP} vectors do not lead to superior results – likely improves the model.” The interpretation here is rather suspect to put it mildly, as $p < 0.5$ does not mean that your null hypothesis is more likely than not to be rejected, it just means that you cannot reject your null hypothesis. It would be more straightforward to just report the absolute values of different performance metrics rather than to worry about assigning a p-value to the difference. This goes for Line 174 and other uses of p-values

to compare model performance – it just feels unnecessary and conceptually problematic.

Response: We recognize that we have previously not done a good job explaining the motivation for the use of p -values, and we apologize for any confusion this may have caused. Our motivation for using p -values was to estimate if different performances between models could be due to chance or point to an inherent superiority of one model over another. They were not meant to be used in selecting the best model, even if our previous writing indicated that. We fully agree with the reviewer that stating the absolute values of model metrics is more important than stating the p -values. However, we still see value in additionally reporting the p -values to indicate to the readers that there is a chance that the chosen model is not indeed better, despite its better performance metrics. This is particularly meaningful in situations where the differences in performance between models are small, as for the comparison between ESM-1b and ESM-1b_ESP vectors.

Action: We reformulated the corresponding paragraph, now focusing mainly on the differences in model performance. Moreover, we clearly motivate why we additionally state p -values (ll. 181ff.):

Since the ESM-1b_{ESP} vectors lead to improved performance on the test set and during CV, we chose to represent enzymes through ESM-1b_{ESP} vectors in the following. However, as model performance is quite similar for both enzyme representations, we tested if the difference in model performance is statistically significant. We used a one-sided Wilcoxon signed-rank test that compared the absolute errors made by both models on the test set, resulting in $p = 0.41$. Thus, the difference in absolute errors is not statistically significant at the commonly used 5% level. This finding indicates that the observed performance improvement with ESM-1b_{ESP} vectors might be due to random effects, and that we cannot be sure that the model with ESM-1b_{ESP} vectors is indeed superior.

7. **(a)** Comparisons of TurNuP to other methods: DLKcat was mentioned and compared quantitatively but a more high level description of the differences in how the methods work would be welcome. **(b)** More seriously, the quantitative comparisons between methods are problematic throughout the manuscripts because the test datasets are different. For example, Lines 215-220 – these are completely meaningless comparisons because the test set is different. If the authors want to compare absolute R^2 values across methods, they need at the very least to test the methods on the same data, of course ensuring that the data does not lie in the training set as well. **(c)** Similarly, the DLKcat comparison again seems to use a different test set if I understand correctly.

Response (a): We agree that the DLKcat model and its differences to TurNuP should be described in more detail.

Action (a): We added additional sentences describing the architecture of the DLKcat model, and we highlighted the differences to our approach (ll. 312ff.):

To predict k_{cat} values, DLKcat uses information extracted from the amino acid sequence and from one of the substrates of the reaction. Instead of using state-of-the-art methods for encoding protein information, i.e., transformer networks applied to protein amino acid sequences, convolutional neural networks (CNNs) were applied. Moreover, since DLKcat only uses information about one of the substrates, important information about additional substrates and the products can be missing.

Response (b): We agree that the scope and the design of Heckmann et al.'s test dataset is very different compared to ours, and thus, we should not compare R^2 values.

Action (b): We now emphasize that we cannot directly compare performance on the test sets between Heckmann et al.'s model and our model (ll. 312 ff.):

Since Heckmann et al.'s test dataset is small and only contains measurements for a subset of reactions from a single organism, it is not possible to directly compare the performance of their model to TurNuP's performance, which was evaluated on a much more extensive and general test dataset.

Response (c): We agree that the R^2 values that DLKcat and TurNuP achieve on their respective test sets are not directly comparable, as the test sets were constructed differently, likely introducing bias. However, we wish to argue that a comparison is indeed appropriate after identifying this bias and correcting for it, given that both models were developed for the same purpose – predicting k_{cat} for the reactions catalyzed by arbitrary enzymes – and were evaluated on large and diverse test data. However, while the DLKcat test set was constructed as a fully random subset of the full data set, the TurNuP test set was constructed such that it contained no enzymes with sequences that are identical to those of the training set, i.e., it is biased towards enzymes that are different from those used for training. To account for this bias, we divided both test sets – those of TurNuP and of DLKcat – into different splits according to the maximal enzyme sequence identity compared to the respective training enzymes. This largely removes the bias caused by the differences in test set construction, and thus comparing the R^2 values of these splits appears appropriate.

Although it is possible that there additional biases exist in the test sets, it is not obvious what such biases would be. Moreover, independent of the existence of such biases that might make the direct comparison of R^2 values difficult, the results clearly highlight that our model can provide meaningful results even for enzymes with low sequence identity (clearly positive R^2), whereas DLKcat fails to make predictions that are better than guessing for those enzymes (clearly negative R^2).

Nevertheless, we acknowledge that we should make it explicit that the comparison of R^2 values on the whole test set does not constitute a direct and fair comparison. Unfortunately, testing both methods on the same test set is not possible. Since both methods used overlapping data to create their training and test splits, test data from one split will be contained in the training

set from the other method. However, we believe that comparing proteome allocation predictions using k_{cat} values from both models does provide a fair comparison.

Action (c): We included an explanation describing that we have to be careful when comparing R^2 values across different test sets, and we now explain that by comparing the R^2 values for the same test splits (i.e., different enzyme sequence identities), we can remove a bias in model performance resulting from different dataset distributions (ll. 323 ff.):

The DLKcat test set was constructed as a random subset of the full data. In comparison, our own test set was constructed such that it contained no enzymes with sequences that are identical to those of the training set, i.e., it is biased towards enzymes that are distinct from those used for training. This intentional bias, which was introduced to assess the ability of TurNuP to extrapolate to new enzymes, means that a direct performance comparison between DLKcat and TurNuP on their respective test sets is not meaningful. To account for this bias, we divided both test sets – those of TurNuP and of DLKcat – into different splits according to the maximal enzyme sequence identity compared to the respective training enzymes (Figure S2). Given that both models were developed for the same purpose — predicting k_{cat} for the reactions catalyzed by arbitrary enzymes — and were evaluated on large and diverse test data, a performance comparison on these sequence identity splits is meaningful.

8. Global underprediction of high k_{cat} s: The distribution in figure 4 looks like their model underpredicts higher k_{cat} s substantially – a 1,1 line should be drawn on this figure as well as a correlation line. Is there any explanation for what seems to be a systemic bias in the model?

Response: We agree that the addition of a diagonal line and a correlation line to Figure 4 is useful. Our model, TurNuP, indeed exhibits limitations in predicting more extreme values, as it underestimates high values and overestimates low values on the test set. This issue is likely due to a common phenomenon known as "regression dilution". Regression dilution occurs when noise is present in the input features, e.g., due to measurement errors or random noise, leading to a regression line with a slope closer to zero than the true regression. As the input features in TurNuP only explain 44% of the variance in k_{cat} , they likely only approximately capture the true determinants of k_{cat} . They can thus be considered noisy representations of those true determinants, leading to regression dilution.

We further hypothesize that non-optimal reaction conditions (such as pH, temperature, missing metal ions) might also play a factor in cases where very low k_{cat} values were observed but higher k_{cat} values were predicted. This hypothesis is supported by analyzing the similarity between different measurements for the same enzyme-reaction pair, showing that especially for k_{cat} values with extremely low measurements, the variance across multiple measurements is high (see Supplementary Figure S6). Unfortunately, we could not include measurement conditions as input features, since they were not available for most data points in the enzyme databases used.

Action: We added a grey dashed diagonal line and a red dashed line displaying the correlation between the predicted and measured k_{cat} values (see Figure 4). We added an additional paragraph to the corresponding Results section aiming to explain why the model might have a systematic bias (ll. 222 ff.):

Figure 4 shows that TurNuP tends to systematically overestimate extremely low and underestimate very high k_{cat} values. This phenomenon likely arises due to a common statistical effect known as regression dilution: noise present in the input features leads to a flattening of the slope of the line that describes the correlation between the predicted and true values for the target variable³⁵. While the chosen input features in TurNuP can account for ~44% of the variance in k_{cat} , they only approximately capture the true, unknown determinants of k_{cat} . The input features can thus be considered noisy representations of those true determinants, leading to regression dilution. We further hypothesize that non-optimal experimental conditions (such as pH, temperature, missing metal ions) play an important factor in cases where very low k_{cat} values were observed but higher k_{cat} values were predicted. Figure S4, which displays the similarity between different measurements for the same enzyme-reaction pairs, supports this hypothesis: a large variance is present especially for very low k_{cat} measurements. Unfortunately, we could not include reaction conditions as input features to overcome this issue, since experimental conditions are not available for most data points in the enzyme databases used.

Minor Comments

9. **(a)** Alternate explored features: The section exploring flux, CAI, and Km as alternative features is weak and probably deserved to be relegated to the discussion. The methods for estimating these parameters are not well tested and subject to their own problems. For example, the use of Kms as a single data point for all reactants during prediction seems greatly flawed and in serious need of validation studies. Error in estimation of these parameters could easily be the sole cause of their failure to aid model predictions, regardless of whether the parameters do have a relationship with k_{cat} . Including the existing results is not problematic per se, but I am worried that readers would confuse a lack of success of these estimated features as implemented to indicate that these features do not relate to k_{cat} . **(b)** For example, I do not believe they have sufficient evidence to make the statement starting on line 360 in the Discussion.

Response (a): We admit that the calculations of fluxes and KM values have flaws. However, we argue that even though we might not be able to create highly accurate values for reaction fluxes and KM values, it is still reasonable that these input features might improve model performance (as seen in a previous approach by Heckmann et al.). It might be an interesting and surprising result for many readers that although those input features are correlated with k_{cat} , using them as additional model input does not improve prediction quality. This indicates that the information provided by the additional input features is already present in the reaction fingerprint and enzyme

representation. Nevertheless, we agree with the reviewer that we should point out the flaws of the calculated input features more clearly and explain that exact values of KM and fluxes could still provide important additional information for k_{cat} prediction models.

Action (a): We expanded the corresponding Results section, explaining that the lack of model improvement does not mean that accurate KM values and fluxes could not improve model performance (ll. 410ff.):

This finding does not preclude the usefulness of flux and KM for predicting k_{cat} . On the one hand, the result suggests that the enzyme and reaction representations already contain the relevant information offered by the additional input features used here. On the other hand, it is possible that accurate measurements of flux and KM, as opposed to the approximate predictions employed here, could enhance the predictive power of these variables for k_{cat} .

Response (b): We recognize that due to the noise in the calculated additional input features, there is indeed not enough evidence to formulate the statement in line 360.

Action (b): We removed the statement from the Discussion.

10. Another prominent approach to k_{cat} estimation is through in vivo k_{cat} estimation using proteomics data. The authors cite the originating paper by Davidi et al but only in another context. Some discussion of this alternate approach and differences in achieved accuracy would be appropriate.

Response: We agree that we should discuss Davidi et al.'s approach for estimating k_{cat} values among other previous k_{cat} prediction methods.

Action: We expanded the Introduction, now also describing Davidi et al.'s approach when summarizing previous k_{cat} prediction methods (ll. 22ff.):

Davidi et al. estimated k_{cat} values for enzymes in Escherichia coli using computationally calculated reaction fluxes and proteomic measurements across 31 different growth conditions. Their approach leads to k_{cat} estimates that are highly correlated with experimentally measured values ($r^2 = 0.62$ on log10-scale). However, this approach is limited to very-well studied organisms, and even for E. coli, k_{cat} values for only 436 enzymes could be estimated in this way.

11. I didn't understand the description in the methods of how they compared DLKcat to TurNuP, specifically these lines from 482-486. Could the authors expand on this explanation or provide a figure on the methods?

Response: We thank the reviewer for drawing our attention to the fact that this description might be insufficient to understand the comparison between DLKcat and TurNuP on different test splits.

Action: We created an additional Figure (Supplementary Figure S2) that displays the process of dividing the test sets into different splits according to maximal sequence identity compared to the training enzymes, and we added explanatory text to the corresponding Methods section (ll. 665ff):

We then grouped the data points in each of the two test sets according to this maximal sequence identity: < 40% ; ≥ 40% and < 80% ; ≥ 80% and < 99% ; and ≥ 99%. For each of the two methods, we then calculated the coefficient of determination R^2 separately in each of the sequence identity groups. For each level of sequence identity, we compared the R^2 values of TurNuP (evaluated on its corresponding subset of test data points) and DLKcat (evaluated on its corresponding subset of test data points).

12. How are reaction conditions (temperature) that could systematically affect k_{cat} values handled? What is the intended temperature of the predictions? 25C?

Response: Reaction conditions can indeed have an important impact on the resulting k_{cat} values. However, as reaction conditions (such as temperature and pH) were not available for the majority of experimentally-measured k_{cat} values in the used enzyme databases, we could not use this information as input features for our prediction model. Checking some of the sources for k_{cat} values showed that most measurements were performed at 22, 25, or 37 degree celsius. The Discussion section already contains a paragraph describing this limitation of TurNuP (ll. 446 ff.).

Action: None

13. No Supplementary dataset? Including some of their data in tables would improve the use of their work by the community, for example flux predictions, CAI values, etc.

Response: All data that was used or calculated to create the results displayed in the manuscript, including flux predictions, CAI values, and Km values, are uploaded to our data repository.

Action: We changed how we store all relevant datasets. We now uploaded all data to Zenodo, which we hope will facilitate easier access. In the manuscript we state where all data can be downloaded (ll. 522ff.; see also our response to comment 1**(b)**):

All datasets used to create the results of this manuscript are available from <https://doi.org/10.5281/zenodo.775531>.

14. Supplementary plots should show best fit line and correlation if the intention is to show a relationship or lack thereof between variables.

Response: We agree that a correlation line helps to interpret the results displayed in these scatter plots.

Action: We added red dashed correlation lines to Supplementary Figures S5, S6, and S7.

15. Figure S1 on normal distributions – why not show a Q-Q plot that is actually intended to show normality, comparing to a normal distribution with the same mean and variance as the observed distribution? Not that this particularly matters – the log transformation is certainly a good thing for model training.

Response: We thank the reviewer for pointing out that Q-Q plots might be more appropriate for comparing the distributions to a normal distribution instead of simply plotting histograms of the empirical distributions.

Action: We replaced the plots in Figure S1 with Q-Q plots.

16. Their website (<https://turnup-cs.hhu.de/>) has a typo in its title ('Turnover Number Prediction . TNP') and when you run a job, the message has a typo "Your result is currently being calculted."

Response: We apologize for the typos on our webpage.

Action: We carefully revised the whole webpage to remove all typos.

17. Around line 68, some Figure formatting ("Figure ??") (recurring issue, maybe with the Supplementary Figures)

Action: We carefully revised the whole manuscript, removing all formatting errors.

Reviewer #2 (Remarks to the Author):

Kroll et al train a deep-learning-based network to predict k_{cat} values from a representation of the substrate or the protein or their combination. Accurate k_{cat} prediction would be valuable for metagenomic screening and enzyme engineering studies, but I see significant problems with the current submission.

Response: We thank the reviewer for providing helpful comments. We appreciate that the reviewer pointed us to areas for improvement, especially with respect to aspects of the manuscripts that we have previously not made clear enough. The comments have been very valuable for helping us ensure that future readers will fully understand our work.

Major comments:

1. It is not clear to me how well the data were separated into training and test sets and it seems that the criteria were rather loose. From page 4, I'm led to believe that only precisely the same amino acid sequence would be excluded, but it is clear that if a method were trained on data on a homolog then it has a very good prior for estimating the activity of the query. I'm not sure that there's a good way out of this conundrum since generating a truly nonredundant set (<35% sequence identity) is likely to lead to a very small training set.

Response: It is true that we separated the training and the set using the loose criteria of not allowing identical amino acid sequences in the training and the test set. However, we show (Figure 5a) that only very few proteins in the test set (~2.5%) are close homologs compared to proteins in the training set. Moreover, we performed a post-hoc analysis, where we divided the test set into different splits according to the sequence identity levels compared to sequences in the training set. This includes a split consisting of 474 test data points with enzyme sequences that have <40% sequence identity compared to sequences in the training set (Figure 5a), which is close to the 35% sequence identity suggested by the reviewer and provides approximately the same insights as directly creating the test set using such a low sequence identity threshold. This analysis shows on the one hand that more than 50% of the sequences in the test set have a maximal sequence identity <40% compared to training sequences, and on the other hand, it shows that TurNuP can still provide good predictions for this subset of data points. We apologize for not explaining well enough that our post-hoc analysis provides insights for how model performance changes for different sequence identity thresholds.

Action: We now state more clearly that for practical reasons, training and test split were created using only a loose criterion, which does not necessarily exclude close homologs in the test set. We describe in more detail that the post-hoc analysis of dividing the test set into different splits regarding sequence identity compared to training proteins provides similar insights as directly splitting the test set using a lower sequence identity threshold (ll. 76ff.):

This splitting procedure does not prevent the inclusion of enzymes in the test set with close homologs in the training set. However, we found that only very few

enzymes in the test set are highly similar to enzymes in the training set, i.e., only 70 out of 850 enzymes in the test set show $\geq 90\%$ sequence identity to enzymes in the training set, and only 26 enzymes show $\geq 98\%$ identity. Moreover, to evaluate model performance for different levels of enzyme similarities (see below), we divided the test set further into subsets with different levels of maximal sequence identities between training and test enzymes (Figure S2).

2. On pg. 5, the authors demonstrate that training on the substrate alone leads to a high success rate in prediction. They don't draw conclusions from this observation (in general, the manuscript is very narrowly focused on the statistical analysis and provides little in biochemical insight). This may imply that, similar to the above point, the method is in effect learning to predict reactivity based on homology, which is a trivial outcome of the training process.

Response: We agree that it is indeed interesting to discuss that using reaction information alone leads to a high accuracy. We also agree with the reviewer that from our previous manuscript it was not clear enough if the model only learns to make good predictions for data points similar to training data points or if it is capable of extrapolating and making good predictions for unseen reactions and not highly similar enzymes.

Action: We now discuss in the Discussion section that the high accuracy resulting from using only reaction information suggests that properties of the reaction have a strong influence on the turnover number, independent of the catalyzing enzyme (ll.482ff.):

Perhaps surprisingly, k_{cat} predictions that are solely based on reaction information – without information on the catalyzing enzyme – already lead to a high coefficient of determination, $R^2 = 0.38$ (Figure 3). This result suggests that properties of the reaction have a strong influence on the turnover number achievable by natural selection on the catalyzing enzyme. Adding enzyme information additionally to the reaction information as model input had only a moderate effect on model performance, indicating that the information for predicting k_{cat} that is stored in the enzyme and in the chemical reaction are strongly overlapping.

As described in the Action above, we now make it more clear that the model generalizes well even to unseen enzymes with low sequence identity compared to training protein sequences. Moreover, to investigate if the prediction model is also capable of generalizing well to unseen chemical reactions, we added a subsection in the Results section showing that TurNuP can generalize to reactions that have not been part of the training set (ll. 277 ff.):

In the previous subsection, we showed that model performance is highest for enzymes that are similar to proteins in the training set. Similarly, it appears likely that the model performs better when making predictions for chemical reactions that are also in the training set. To test this hypothesis, we divided the test set into data points with reactions that occurred in the training set ($N = 354$) and those with

reactions that did not occur in the training set ($N = 496$).

Unsurprisingly, TurNuP performs better for those data points with reactions that occurred in the training set, achieving a coefficient of determination $R^2 = 0.57$, a mean squared error $MSE = 0.51$, and a Pearson correlation coefficient $r = 0.78$ (Figure 6). However, for those test data points with reactions that TurNuP has not seen during training, model performance is still good, resulting in a coefficient of determination $R^2 = 0.35$, mean squared error $MSE = 1.02$, and Pearson correlation coefficient $r = 0.60$.

For those test data points with reactions not present in the training set, we wondered if a high similarity of the reaction compared to at least one reaction in the training set leads to improved predictions, analogous to what we observed for enzymes with higher sequence identities. For each reaction not present in the training set, we calculated a maximal pairwise similarity score compared to all reactions in the training set based on their structural reaction fingerprints. We indeed found that prediction performance is higher for those data points with reactions more related to training reactions (Figure 6). Even for reactions that are not highly similar to training reactions, i.e., reactions with a similarity score between 0.4 and 0.8, prediction accuracy is still moderate with a coefficient of determination $R^2 = 0.20$, a mean squared error $MSE = 1.30$, and a Pearson correlation coefficient $r = 0.47$. Only for those 17 test data points with reactions that share almost no similarity compared to training reactions, model performance is low, achieving a coefficient of determination $R^2 = 0.06$, a mean squared error $MSE = 1.26$, and a Pearson correlation coefficient $r = 0.29$. We conclude that TurNuP can provide useful predictions for most unseen reactions even if these are not highly similar to previously seen reactions.

3. **(a)** The data presented in Fig. 4 correlates predicted versus measured k_{cat} values shows that the correlation is strong in log scale but that there is substantial noise within an order of magnitude. This is perhaps to be expected but the authors should acknowledge this, because in many enzymology studies, a much higher accuracy would be needed. **(b)** Second, it may help their case if they showed that measurements of the same protein in two different studies (where such data exist) are also noisy. I believe this to be the case but I don't know what magnitude of error to expect.

Response (a): We agree that we should emphasize that the correlation is in log-scale and that predictions can only be provided with an accuracy of up to an order of magnitude.

Action (a): We now added an explanation to the corresponding Results section, where we translate the mean deviation on log-scale to non-log-scale to indicate what average accuracy can be expected when using the prediction model (ll. 216ff.):

TurNuP achieves a mean absolute deviation of predicted from experimental k_{cat} values of 0.69 on a log10-scale, which means that predictions and measured values deviate on average by 4.8-fold. Plotting the correlation between different

experimental measurements for the same enzyme-reaction pair (Figure S4) shows that there is substantial variance even between measurements for the same k_{cat} value. This noise in the training and test data indicates that it is difficult to develop a prediction model achieving much better accuracy unless less noisy data becomes available.

Response (b): We thank the reviewer for suggesting to compare experimentally-measured k_{cat} values from two different measurements for the same enzyme-reaction pairs. We believe that this analysis nicely displays why it is not possible to expect a much higher prediction accuracy using the prediction model.

Action (b): We created a scatter plot (similar to Figure 4) that plots experimentally measured k_{cat} for the same enzyme-reaction pairs from two different measurements against each other (ll. 217ff.; Supplementary Figure S4).

4. As a very general comment, the paper is not written well enough in my opinion. There are many typos throughout (even the first line of the Introduction), and it is very lengthy compared to the amount of information and insight that is presented.

Response: We regret that the reviewer feels that the manuscript is not written well enough, and we apologize for the typos.

Action: We carefully revised the whole manuscript to remove all typos.

5. On line 160, the authors seem to acknowledge that they chose their models based on the resulting p -value. This is known as p -value hacking, and if they employed this strategy, they should make it explicit that they used much more stringent p -values than are otherwise common.

Response: While we may not have made this sufficiently clear in the previous manuscript, we did in fact not choose models based on the resulting p -values. Instead, we chose models based on their performance, evaluated through the three metrics mean squared error (MSE), Pearson correlation coefficient (r), and coefficient of determination (R^2). Only after selecting models according to these metrics, we calculated the p -value to quantify our confidence that the chosen model is indeed superior to an alternative model. The section pointed out by the reviewer is an example of this procedure: we selected the model based on the performance metrics, and although the following statistical test did not show that we can conclude that there is a statistically significant difference, we did not change our model selection. However, we acknowledge that in the previous version of the manuscript, we did not make our procedure for model selection and the motivation for calculating p -values clear enough, and we see that our description was prone to cause misunderstandings.

Action: We now make it much clearer which metrics we used to select one model over another.

Moreover, we motivate why we calculated the p -values (ll. 135ff., ll. 180ff.).

As the DRFPs led to improved performance on the test set and during CV (Figure 3), we chose DRFPs to represent the catalyzed chemical reactions in the further analyses. To test if the better performance of the DRFPs is statistically significant, we used a two-sided Wilcoxon signed-rank test that compared the absolute errors of the models on the test set, resulting in $p = 0.064$ (DRFP vs. difference fingerprints) and $p = 2.61 \times 10^{-4}$ (DRFP vs. structural fingerprint). Thus, while the difference in absolute errors between DRFPs and difference reaction fingerprints is not statistically significant at the commonly used 5% level, a value close to 0.05 indicates that the DRFPs indeed likely lead to improved model performance. [...]

Since the ESM-1b_{ESP} vectors lead to improved performance on the test set and during CV, we chose to represent enzymes through ESM-1b_{ESP} vectors in the following. However, as model performance is quite similar for both enzyme representations, we tested if the difference in model performance is statistically significant. We used a one-sided Wilcoxon signed-rank test that compared the absolute errors made by both models on the test set, resulting in $p = 0.41$. Thus, the difference in absolute errors is not statistically significant at the commonly used 5% level. This finding indicates that the observed performance improvement with ESM-1b_{ESP} vectors might be due to random effects, and that we cannot be sure that the model with ESM-1b_{ESP} vectors is indeed superior.

Reviewer #3 (Remarks to the Author):

The topic is well-grounded as an incremental work from Nature Catalysis [16]. The deep transformer learning methodology looks appropriate. The results are minimally enough. The deliverables are very nice, including website, source code, and clearly reproducible procedure details in the writing. Taking into the above, I guess the submission could be revised. My comments are as follows:

Response: We sincerely appreciate the positive and thoughtful assessment of our manuscript. The reviewer's feedback pointed out important aspects that we had not previously explored or considered, such as performing a comparison to additional machine and deep learning models, facilitating an easier use of our web server through provided input examples, or focusing on the biological insights that can be gained from our prediction model.

Major Comments:

1. **(a)** The only competing method to be compared is DLKcat which is published on Nature Catalysis [16]

[16] Li, F. et al. Deep learning-based k_{cat} prediction enables improved enzyme-constrained model reconstruction. Nat. Catal., 1–11 (2022)

I was wondering if other possible approaches should be compared; for instance, other machine learning and deep learning approaches? **(b)** Baseline estimations are needed on top of the comparisons to [16].

Response (a): As pointed out by the reviewer, DLKcat is the only other general method for predicting k_{cat} , and thus, it is the only model which we can use for a more direct comparison. However, we agree that a comparison to additional deep and machine learning approaches would strengthen our choice of machine learning algorithm.

Action (a): We performed hyperparameter optimizations for three additional machine and deep learning models (linear regression, random forest, and neural network) to justify our choice of machine learning algorithm (ll. 235ff.; method details in ll. 649ff.):

To compare the gradient boosting model to alternative machine learning models, we also trained a linear regression model, a random forest model, and a fully connected neural network for the task of predicting k_{cat} values from the combined ESM-1bESP and DRFP vectors. However, these models performed worse compared to the gradient boosting model (Supplementary Table 1). To test if the better performance of the gradient boosting model is statistically significant, we used a one-sided Wilcoxon signed-rank test that compared the absolute errors of the models on the test set, resulting in $p = 4.29 \times 10^{-11}$ (gradient boosting vs. linear regression), $p = 3.38 \times 10^{-7}$ (gradient boosting vs. random forest), and $p = 8.97 \times 10^{-4}$ (gradient boosting vs. neural network). Thus, the additional machine learning

models indeed lead to statistically significant worse results compared to the gradient boosting model.

[...]

To compare the performance of the gradient boosting model to additional machine learning models, we also trained a linear regression model, a random forest model, and a fully connected neural network for the same prediction task. To find the best hyperparameters for the models, we again performed 5-fold CVs on the training set. For the random forest model, the optimized hyperparameters were the number of estimators, maximal depth of trees, and minimum samples per leaf. For the linear regression model, we searched for the best L1 and L2 regularization coefficients. We used the Python package scikit-learn for training both models. For the fully connected neural network, the optimized hyperparameters were learning rate, learning rate decay and momentum, dimension of hidden layers, batch size, L2 regularization coefficient, and number of epochs. We used the deep learning library TensorFlow to train the fully connected neural networks.

Response (b): We already included a comparison to a baseline estimation in our manuscript (II. 263ff.): We compared our approach to the simple approach of calculating the mean k_{cat} value of the three most similar enzymes with experimentally-measured k_{cat} values and using the resulting value as a prediction for the k_{cat} value. We show that our model clearly outperforms this simple baseline.

Action (b): We make it more clear that we also compare our model to a baseline approach, which we now already point out in the Introduction (I. 50f.):

Our resulting Turnover Number Prediction model – TurNuP – outperforms a simple similarity-based approach and both previous methods for predicting k_{cat} .

2. The authors heavily focused on performance gains (i.e. minimizing errors). Assuming the proposed method is fine, would there be new / novel insight into the problem domain (i.e. enzymatic turnovers)?

Response: Unfortunately, gaining insights into the importance of specific reaction or enzyme properties is not straightforward. The trained gradient boosting model could be used to calculate the importance of every dimension of the input feature vectors, but interpreting the meaning of single input feature dimensions is hardly possible. This is because the ESM-1b vectors were created by iteratively extracting information from the amino acid sequence using a deep neural network with 33 layers, and the reaction fingerprints were created from large binary reactant fingerprints, where the meaning of each entry cannot be easily interpreted. However, our model shows that much of the variance of the k_{cat} values can be explained by only using information about the chemical reaction or by only using information about the catalyzing enzyme. Thus, our results indicate that the information for predicting k_{cat} that is stored in the enzyme and in the chemical reaction are strongly overlapping.

Action: We added a paragraph to the discussion section, discussing why it is difficult to extract biochemical insights from the prediction model (ll. 488ff.):

It would be desirable to analyze which properties of the chemical reactions and of the enzymes are most relevant for predicting k_{cat} . Unfortunately, because of how we encoded the input information, it is difficult to draw conclusions about the importance of certain properties of the reactions and enzymes from analyzing the importance of the input features. For example, we could extract which input features from the ESM-1b vectors are most relevant for predicting k_{cat} , but we cannot easily relate entries in the ESM-1b vectors to properties of the proteins, as the ESM-1b vectors were created in a previous step by iteratively extracting information from the amino acid sequence using a deep neural network with 33 layers.

3. Deep learning is known to be slow. Running time analysis could be depicted with the hardware setup details.

Response: We agree that providing a running time analysis provides valuable insights for further research and for reproducing the results of the manuscript.

Action: We expanded the Methods section, now providing details about the used hardware and also providing information about run-times for training and hyperparameter optimizations (ll. 740ff.):

To perform hyperparameter optimization for all gradient boosting models, we used the High Performance Computing Cluster at the University of Düsseldorf (Germany). Each hyperparameter optimization was executed for 48 hours on an Nvidia Quadro RTX 8000, which resulted in testing $\sim 5\,000$ different hyperparameter settings for each model.

4. Deep learning methodology reviewers may argue the neural network architecture should be visualized and examined for further details with ablation analysis. I am flexible with it though.

Response: We agree that further architecture details, in particular for the calculation of the ESM-1b_ESP-vectors, is desirable.

Action: We further expanded the methods section, explaining the architecture of the ESM-1b_ESP-model in much more detail (ll. 594ff.):

As an alternative to the original ESM-1b enzyme representations, we employed fine-tuned enzyme representations that we created previously for the task of predicting the substrate scope of enzymes. These ESM-1b_{ESP} vectors were constructed using code and models provided on the following GitHub repository:

<https://github.com/AlexanderKroll/ESP>.

We here provide a short summary of the construction. First, the original ESM-1b model was slightly modified, i.e., additionally to the input tokens for every amino acid in a protein sequence, an extra token for the whole enzyme was added. This token does not add any additional input information to the model. Instead, this token is used to store information about the whole input sequence that is salient to the downstream task. During training, all tokens, including the extra enzyme token, are first updated iteratively for 33 times, as described above for the original ESM-1b model. Next, the updated additional enzyme token was concatenated with a 1024 dimensional expert-crafted binary representation of a small molecule, the extended-connectivity fingerprint (ECFP). This concatenated vector was used as the input for a fully connected neural network that was trained to predict if the small molecule is a substrate for the given enzyme. The whole model, the ESM-1b and the fully connected layers, were trained end-to-end for the task of identifying substrates for enzymes (Figure S3b). The training set consisted of 287386 positive enzyme-substrate pairs with either phylogenetically inferred or experimental evidence extracted from the GO Annotation database and with randomly sampled negative enzyme-small molecule pairs; note that phylogenetic evidence was only used to construct the ESM-1b model, but was not used to train or evaluate the final TurNuP model. During the training process, the model was forced to extract all relevant information for the prediction task from the amino acid sequence and store it in the updated enzyme token. This trained model was used to create the fine-tuned enzyme ESM-1bESP vectors for every enzyme in our dataset by extracting the updated enzyme representation from the model.

We now also provide a Supplementary Figure (Figure S3) that visualizes the architecture of the used ESM-1b Transformer Networks.

5. Statistical significance testing could be conducted; for instance, p-value calculations across multiple runs of cross-validations.

Response: We already conducted statistical testing for analyzing if the difference in model performance between different models is statistically significant (ll. 136ff., ll. 175ff., ll 207ff., ll. 272ff., ll. 334ff., ll. 351ff., and ll. 678ff.). We performed these tests by comparing the absolute errors of the models on the test set.

Action: We now additionally test if there are statistically significant differences between the alternative machine learning models implemented (ll.184.ff):

We used a one-sided Wilcoxon signed-rank test that compared the absolute errors made by both models on the test set, resulting in $p = 0.41$. Thus, the difference in absolute errors is not statistically significant at the commonly used 5% level. This finding indicates that the observed performance improvement with ESM-1b_{ESP} vectors might be due to random effects, and that we cannot be sure that the model

with ESM-1b_{ESP} vectors is indeed superior.

Minor Comments:

6. I can see there are formatting errors " (Figure ??). " on page 14/30.

Response: We apologize for these formatting errors.

Action: We carefully revised the whole manuscript, removing all such errors.

7. An example sequence should be given in the webpage for user-friendly usage.

Response: We agree that an example input will support an easy use of our web server.

Action: We now provide an example input for both functionalities of the web server, i.e., the prediction of single enzymatic turnover numbers using the online form as well as for the prediction of multiple k_{cat} values, which requires the upload of a csv-file.

8. **(a)** I cannot find the complete source code download on the webpage, but on the GitHub link provided in the Methods section. **(b)** Standalone executables on Linux and Windows could also be provided.

Response: (a) Previously, we did not provide the source code on the webpage. We agree that it is helpful to provide a link to the GitHub repository containing the source code of the prediction model.

Action: (a) We now provide links on the webpage to the GitHub repositories containing the source code of the manuscript and to an easy-usable Python function for the prediction model.

Response (b): The web server allows users to use our prediction model on any system without the requirement of installing additional software and without executing any computations on the user's PC. Moreover, in addition to the source code of our manuscript (https://github.com/AlexanderKroll/kcat_prediction), we also provide an easy-usable Python function for our prediction model (https://github.com/AlexanderKroll/kcat_prediction_function), which can be executed on any system. Thus, we believe that providing standalone executables would not provide substantial additional value.

Action (b): None

9. The webpage looks very simple and does not contain enough documentations and tutorials.

Response: We agree that more documentation is desirable for the web server.

Action: As described above in our Action for comment 7 of this reviewer, we now provide example inputs for both functionalities of the web server. Moreover, we updated the help page of the web server, explaining the model output and the input types that are suitable for the prediction model.

REVIEWERS' COMMENTS

Reviewer #1 (Remarks to the Author):

The authors thoroughly addressed my previous concerns. While it is disappointing that model interpretation could not be addressed in this work, I hope that a future focused work can develop this topic. I have no further issues

Reviewer #2 (Remarks to the Author):

The authors have addressed my comments.

Reviewer #3 (Remarks to the Author):

The authors have addressed my comments comprehensively.